# Coordinated regulation of the entry and exit steps of aromatic amino acid biosynthesis supports the dual lignin pathway in grasses

Jorge El-Azaz[1], Bethany Moore[1,2], Yuri Takeda-Kimura[1,3], Ryo Yokoyama[1,4], Micha Wijesingha Ahchige [1,4], Xuan Chen [1,5], Matthew Schneider[1,6] & Hiroshi A. Maeda [1] ✉

Vascular plants direct large amounts of carbon to produce the aromatic amino acid phenylalanine to support the production of lignin and other phenylpropanoids. Uniquely, grasses, which include many major crops, can synthesize lignin and phenylpropanoids from both phenylalanine and tyrosine. However, how grasses regulate aromatic amino acid biosynthesis to feed this dual lignin pathway is unknown. Here we show, by stable-isotope labeling, that grasses produce tyrosine >10-times faster than Arabidopsis without compromising phenylalanine biosynthesis. Detailed in vitro enzyme characterization and combinatorial *in planta* expression uncovered that coordinated expression of specific enzyme isoforms at the entry and exit steps of the aromatic amino acid pathway enables grasses to maintain high production of both tyrosine and phenylalanine, the precursors of the dual lignin pathway. These findings highlight the complex regulation of plant aromatic amino acid biosynthesis and provide novel genetic tools to engineer the interface of primary and specialized metabolism in plants.

The biosynthesis of aromatic amino acids (AAAs)—phenylalanine, tyrosine, and tryptophan—represents one of the major routes of plant metabolism that supplies essential building blocks for the production of proteins and numerous plant natural products[1,2]. Yet, it remains poorly understood how the AAA biosynthetic pathway is regulated to meet various demands for AAA precursors in different species. The most abundant of these AAA derived compounds is lignin, which accounts for up to 30% of plant dry weight[3] and plays a critical role in strengthening and waterproofing secondary cell walls. In most plant species, lignin and other phenylpropanoids are synthesized exclusively from phenylalanine by the enzyme phenylalanine ammonia-lyase (PAL, Fig. 1a)[4,5]. Conversely, grasses (family Poaceae), arguably one of the most important plant lineages from both an ecological and economic perspective, can produce lignin from both tyrosine and

phenylalanine due to the presence of a bifunctional phenylalanine/tyrosine ammonia-lyase (PTAL) enzyme(s)[6–11]. Multiple lines of evidence support that a significant proportion of grass lignin is synthesized from tyrosine[10,12,13]. However, it remains unknown how grasses regulate the upstream AAA biosynthetic pathways to provide high amounts of both tyrosine and phenylalanine precursors to support the unique dual lignin pathway.

Our current knowledge on the regulation of plant AAA biosynthesis, mostly derived from dicot models, indicates that plants balance AAA production by targeting activities of key enzymes of the AAA pathway(s) through a combination of transcriptional and feedback regulation[1,2,14]. For instance, the first enzyme in AAA biosynthesis, 3-deoxy-D-*arabino*-heptulosonate 7-phosphate (DAHP) synthase (DHS; EC:2.5.1.54) (Fig. 1a), is feedback regulated by AAAs and multiple

[1]Department of Botany, University of Wisconsin-Madison, Madison, WI, USA. [2]Present address: Morgridge Institute for Research, Madison, WI, USA. [3]Present address: Faculty of Agriculture, Yamagata University, Yamagata-shi, Japan. [4]Present address: Max Planck Institute of Molecular Plant Physiology, Potsdam-Golm, Germany. [5]Present address: International Institute of Tea Industry Innovation for "one Belt, one Road", Nanjing Agricultural University, Nanjing, Jiangsu, PR China. [6]Present address: Cell Culture Company, Minneapolis, MN, USA. ✉e-mail: maeda2@wisc.edu

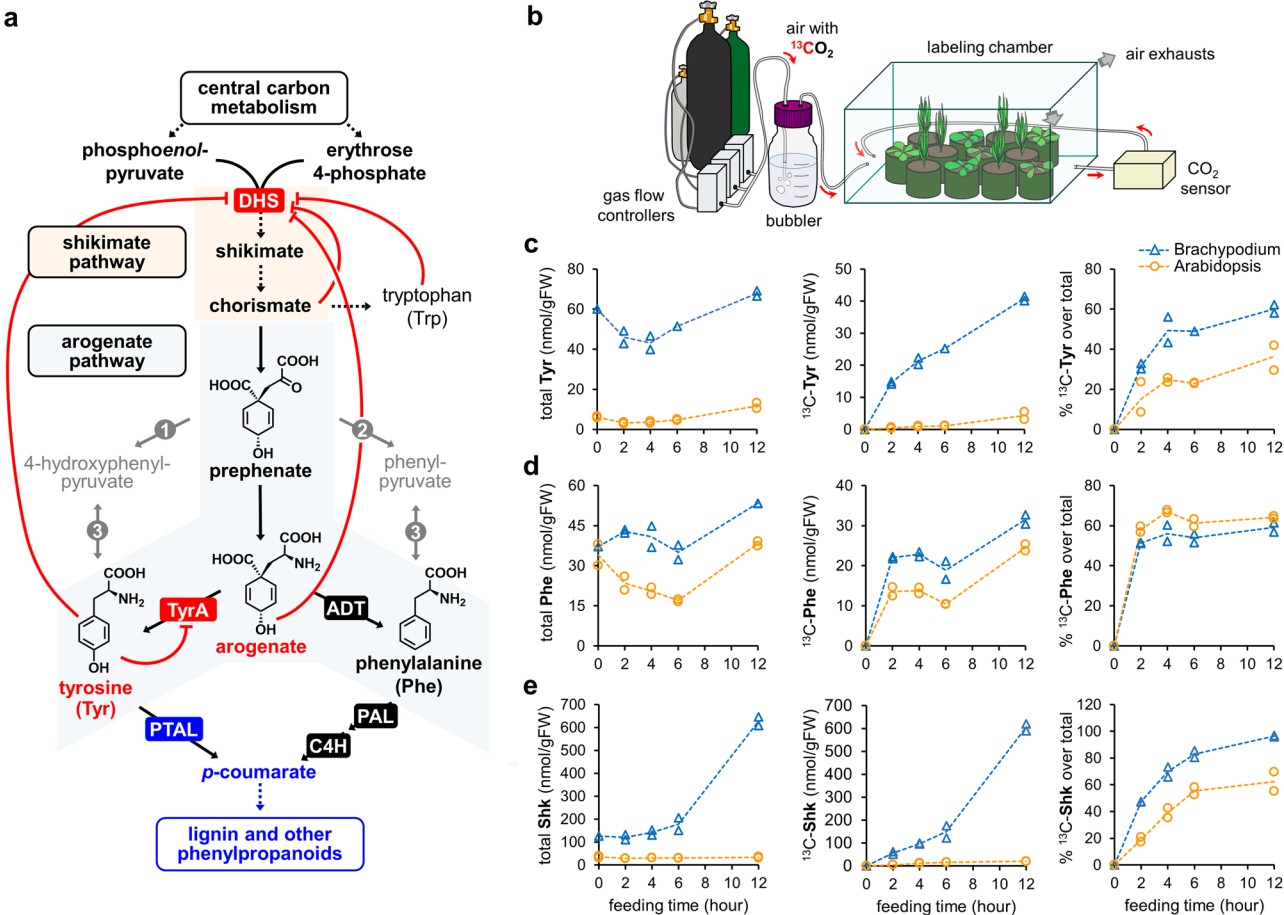

**Fig. 1 | $^{13}CO_2$ feeding uncovers a highly active tyrosine biosynthesis in *Brachypodium distachyon*. a** Plant pre- and post-chorismate aromatic amino acid biosynthesis pathways and known targets for feedback regulation (red lines). The grass-unique bifunctional phenylalanine tyrosine ammonia-lyase (PTAL, in blue) introduces a shortcut in the phenylpropanoid pathway, transforming tyrosine into *p*-coumaric acid in a single step. Dashed arrows indicate multiple enzymatic steps. Enzyme's abbreviations: DHS, 3-Deoxy-D-*arabino*-heptulosonate 7-phosphate synthase; TyrA, arogenate dehydrogenase; ADT, arogenate dehydratase; PTAL, phenylalanine/tyrosine ammonia-lyase; PAL, phenylalanine ammonia-lyase; C4H, cinnamate 4-hydroxylase; 1, prephenate dehydrogenase TyrA$_p$ (EC 4.2.1.91), only found in legumes; 2, prephenate dehydratase, a side activity of plant ADTs; 3, aromatic amino acid aminotransferase. **b** A schematic representation of the $^{13}CO_2$ feeding circuit. **c**–**e** Total content per gram of fresh weight (left panel), the content of $^{13}$C-labeled metabolite (central panel), and relative $^{13}$C-labeled metabolite over the total (right panel) for **c** tyrosine, **d** phenylalanine, and **e** shikimate, comparing 4-weeks-old *Brachypodium distachyon* (blue triangles) and *Arabidopsis thaliana* (orange circles) plants. Individual datapoints correspond to independent plants sampled at that time point. Dashed lines connect the average value of each time point. Source data for panels **c** to **e** are provided within the Source Data File.

---

downstream metabolites[15,16]. This feedback regulation at the entry point of the shikimate pathway is key to control the biosynthesis of AAAs and, when released by point mutations targeting the regulatory domain of DHS, can largely increase AAA production and $CO_2$ fixation in *Arabidopsis thaliana*[16]. Similarly, the key enzymes controlling tyrosine and phenylalanine biosynthesis from arogenate, arogenate dehydrogenase (TyrA; EC 1.3.1.78) and arogenate dehydratase (ADT; EC 4.2.1.91), are subjected to feedback inhibition by their corresponding reaction products (Fig. 1a)[2,14]. However, while TyrA enzymes are sensitive to feedback inhibition even at low tyrosine levels[17–20], vascular plants have specialized ADT isoforms that maintain their activity in the presence of high concentrations of phenylalanine[21,22]. Furthermore, unlike bacterial DHSs[23], none of the plant DHS enzymes characterized so far are inhibited by phenylalanine[15,24], likely providing abundant phenylalanine precursor for phenylpropanoid production. Besides the feedback regulation at the enzyme level, *DHS* and *ADT* genes are often strongly co-expressed with *PAL* and other lignin and phenylpropanoid-related genes across different plants[15,25–29]. Through this conjunction of transcriptional and feedback regulation, many plant species prioritize the production of phenylalanine for phenylpropanoid biosynthesis, often at the expense of tyrosine and

tryptophan levels[26,30–33]. However, given the presence of the unique dual lignin pathway, we hypothesized that grasses may regulate AAA production differently from other plants.

Here, we combined stable-isotope labeling, phylogenetic and expression analyses, detailed enzyme characterization, and combinatorial *in planta* expression to uncover that the coordinated regulation of the entry and final steps of AAA biosynthesis allows grasses to efficiently provide both tyrosine and phenylalanine precursors to meet the unique demand of the dual tyrosine/phenylalanine lignin pathway. This study highlights the importance of transcriptional and biochemical regulation at key metabolic branching points in fine-tuning the supply of AAA precursors for the downstream lignin and phenylpropanoid pathway. These basic findings and the novel enzymes identified in grasses can be utilized to engineer plants to efficiently produce natural and bio-based aromatic products in plants.

## Results

### Grasses synthesize tyrosine at a much higher rate than Arabidopsis without compromising phenylalanine production

While prior studies reported that grasses accumulate high levels of tyrosine[34–36], an elevated steady-state level of a metabolite does not

necessarily indicate a high synthesis and usage rate[37]. Therefore, we performed $^{13}CO_2$ feeding experiments to compare the turnover rates of AAAs between the grass *Brachypodium distachyon* Bd21-3 and the dicot *Arabidopsis thaliana* Col-0 (hereafter, Brachypodium and Arabidopsis, respectively). Four-weeks-old Brachypodium and Arabidopsis plants, before bolting, were fed side by side (Fig. 1b) with an air mixture containing ~400 ppm of $^{13}CO_2$. Then, samples were collected at regular intervals for determination of $^{13}C$ labeled tyrosine, phenylalanine and shikimate by ultra-high performance liquid chromatography coupled to electrospray ionization mass spectrometry (UHPLC-MS).

Total tyrosine content (with either $^{12}C$ or $^{13}C$) ranged between ~3 and 12 nmol per gram of fresh weight (nmol/gFW) in Arabidopsis, but was much higher in Brachypodium (Fig. 1c), reaching up to 70 nmol/gFW. Moreover, $^{13}C$-labeled tyrosine (mostly eight or nine $^{13}C$-isotopologues, Supplemental Fig. S1) was 10-times more abundant in Brachypodium (~40 nmol/gFW) than in Arabidopsis (~4 nmol/gFW) after 12 h (Fig. 1c). In contrast, total phenylalanine levels were comparable between the two species, in the range of 35–55 nmol/gFW in Brachypodium, and 20–35 nmol/gFW, in Arabidopsis (Fig. 1d). Labeled $^{13}C$-phenylalanine over time was also similar between the two species (Fig. 1d). As observed for tyrosine, most $^{13}C$-phenylalanine was fully labeled, containing eight or nine $^{13}C$ atoms (Supplemental Fig. S1). In addition, we detected striking differences in the dynamics of the shikimate pool, with up to 20-times more total shikimate accumulating in Brachypodium than in Arabidopsis by the end of the day, and a higher incorporation rate of $^{13}C$ (Fig. 1e).

We next performed additional $^{13}CO_2$ labeling experiments using older six-weeks-old plants of Arabidopsis, Brachypodium, and *Setaria viridis* A10.1 (hereafter, Setaria), comparing young leaves with elongating stems, where lignin is actively formed. These experiments further confirmed that grass species accumulate more tyrosine than Arabidopsis (Fig. 2a) and showed that incorporation of $^{13}C$ into tyrosine was particularly rapid in grass stems, which accumulated up to ~50 nmol/gFW of $^{13}C$-tyrosine after 3 h of $^{13}CO_2$ feeding (Fig. 2a), more than 10-times faster than Arabidopsis ( < 2 nmol/gFW; Fig. 2a). On the contrary, the three species exhibited comparable labeling kinetics for phenylalanine, with faster $^{13}C$-phenylalanine accumulation in the stems than in the leaves (Fig. 2b). In the case of shikimate, Arabidopsis stems showed 10 to 20-times more total shikimate and higher rate of $^{13}C$-labeling than the leaves (Fig. 2c), despite high biological variation. In contrast, grass species showed a faster shikimate labeling in the leaves (Fig. 2c). The time-course labeling of different $^{13}C$-isotopologues of phenylalanine, tyrosine, and shikimate differed between leaf and stem tissues and among species (Supplemental Fig. S2). These results showed that, while the three species have a high rate of phenylalanine biosynthesis in the stems, only grasses exhibit high tyrosine turnover in this organ. Furthermore, the high rate of tyrosine biosynthesis in grasses did not seem to compromise phenylalanine biosynthesis.

## Grass TyrA1 and TyrA_{nc} isoforms are highly expressed in growing stems

To understand the mechanism behind the increased production of tyrosine in grasses, we next examined the family of grass TyrA enzymes, which catalyze the final and key regulatory step in tyrosine biosynthesis[2,14,38]. The reconstruction of the plant TyrA protein phylogeny showed that grass genomes have at least three TyrA isoforms (Fig. 3a) corresponding to the *Brachypodium distachyon v3.2* loci Bradi1g34789, Bradi1g34807, and Bradi1g39160, which we named as TyrA1, TyrA2 and non-canonical TyrA (TyrA_{nc}), respectively. Whereas grass TyrA1 and TyrA2 are closely related to each other and to TyrA enzymes from most dicot plants, grass TyrA_{nc} cluster in a more distant group that is sister to cytosolic TyrAnc enzymes from legumes and other dicots (Fig. 3a)[38,39]. All three BdTyrA proteins have predicted plastid transit peptides in their N-terminus (TargetP – 2.0, DTU Health Tech), similarly to most plant TyrA enzymes[39], and were targeted to the plastids when expressed in Arabidopsis protoplast fused to enhanced green fluorescent protein (EGFP) in their C-termini (Supplemental Fig. S3).

To examine the potential involvement of TyrA in the tyrosine-lignin pathway of grasses, we compared the expression of *TyrA* genes with *PTAL* using publicly available expression datasets from

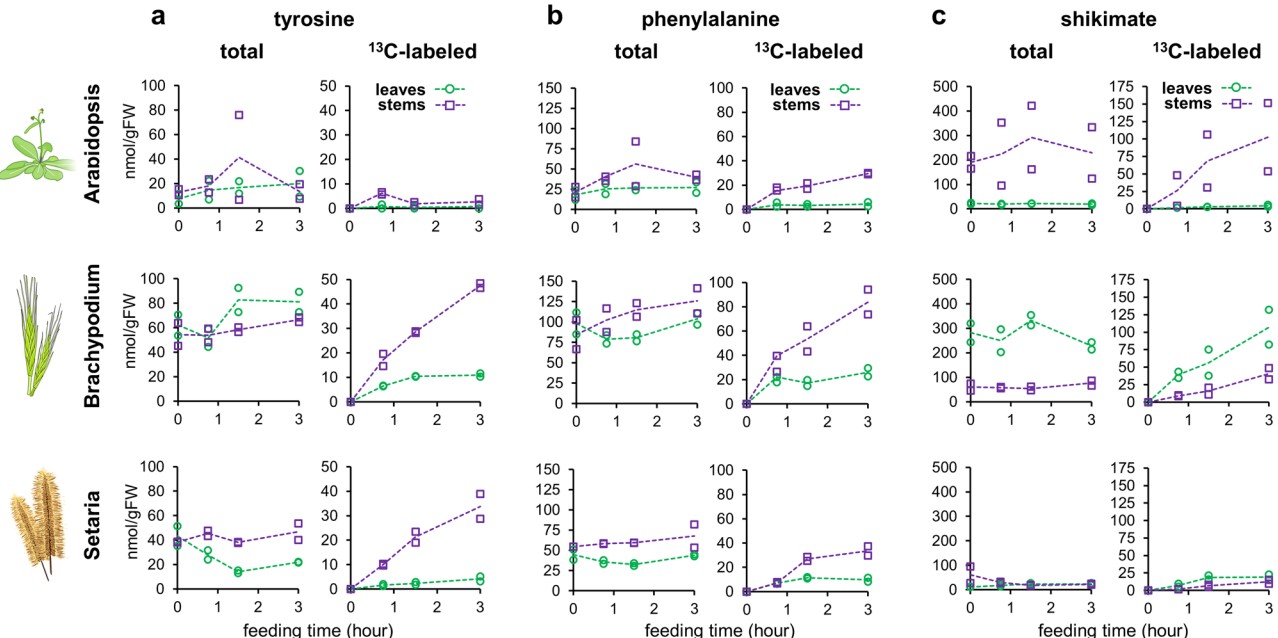

**Fig. 2 | Comparison of $^{13}C$-labeled leaves and stems reveals that grass stems, unlike Arabidopsis, maintain a high rate of both tyrosine and phenylalanine production.** Total and $^{13}C$-labeled **a** tyrosine, **b** phenylalanine, and **c** shikimate determined in leaves (green, circles) and developing stems (purple, squares) of 6-weeks-old *Arabidopsis thaliana* (top), *Brachypodium distachyon* (center), and *Setaria viridis* (bottom) plants. Note that different scales have been used for the individual panels. Individual datapoints corresponds to independent plants sampled at that time point. Dashed lines connect the average value of each time point. Source data are provided within the Source Data File.

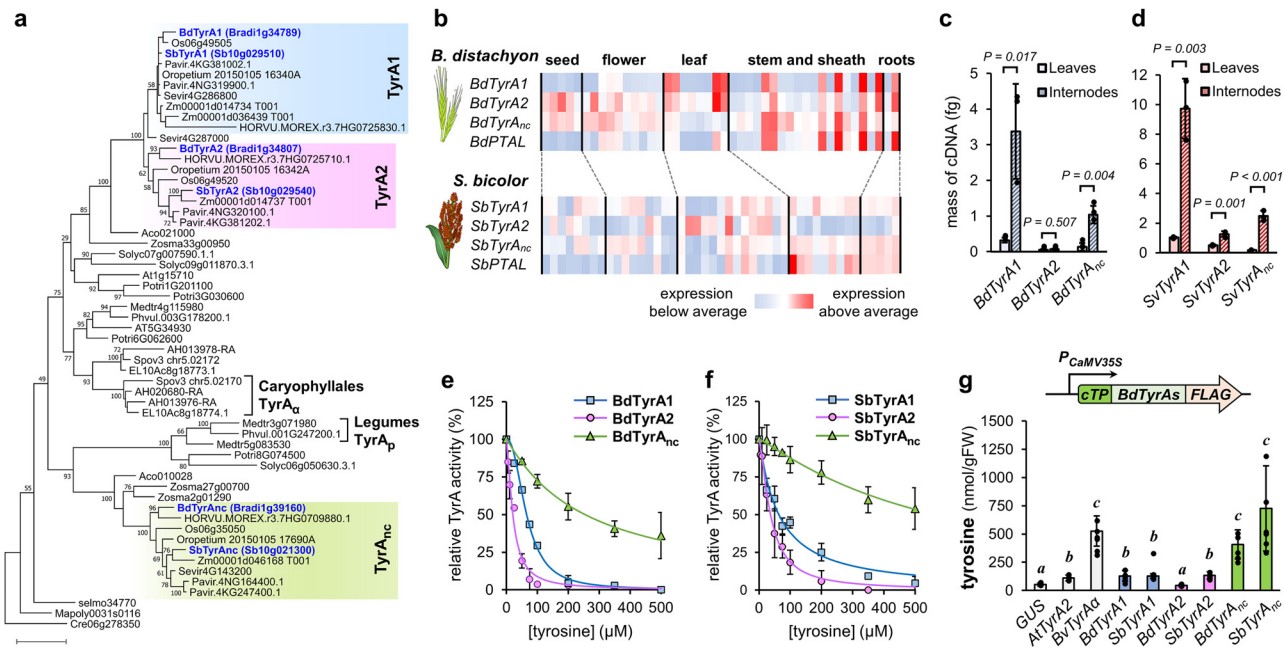

**Fig. 3 | Grass TyrAs differ in their transcriptional and biochemical regulation.** **a** Phylogeny of plant TyrA proteins (outgroup: *Chlamydomonas reinhardtii*) highlighting three clades of grass TyrA1, TyrA2, and non-canonical TyrA (TyrAnc) enzymes. Sequences highlighted in blue have been characterized in this study. Bootstrap test values (based on 1000 replications) below 50 have been omitted. Scale bar indicates number of amino acid substitutions per site. Species abbreviations: AH, *Amaranthus hypochondriacus;* Aco, *Ananas comosus* v3; At, *Arabidopsis thaliana*; EL, *Beta vulgaris* EL10_1.0; Cre, *Chlamydomonas reinhardtii*; Bd, *Brachypodium distachyon*; HORVU, *Hordeum vulgare*; Mapoly, *Marchantia polymorpha*; Medtr, *Medicago truncatula*; Oropetium, *Oropetium thomaeum*, LOC_Os, *Oryza sativa*; Pavir, *Panicum virgatum*; Phvul, *Phaseolus vulgaris*; Potri, *Populus trichocarpa*; selmo, *Selaginella moellendorffii*; Sevir, *Setaria viridis*; Solyc, *Solanum lycopersicum*; Sb, *Sorghum bicolor*; Zm, *Zea mays*; Zosma, *Zostera marina*. **b** Expression patterns of *TyrA* and *PTAL* genes in *Brachypodium distachyon* and *Sorghum bicolor* across different organs and developmental stages. The levels of expression are relative to the average abundance for each individual gene. Data were obtained from PlaNet[40] and MOROKOSHI[41] for Brachypodium and Sorghum, respectively. **c, d** RT-qPCR analysis of *TyrA* transcript abundance in young leaves and internodes from *Brachypodium distachyon* **c** and *Setaria viridis* **d**. Data presented as the average of $n = 3$ individual plants; error bars = *SD*; *P* values according to Student's *t*-test (two-sided test for two samples with equal variance). **e, f** *In vitro* sensitivity to feedback-inhibition by tyrosine of the recombinant TyrA enzymes of Brachypodium **e** and Sorghum **f**. Data points represent the average of at least two separate experiments conducted with different batches of purified recombinant enzyme; error bars = *SD*. **g** Tyrosine content per gram of fresh weight at 72 h following *Agrobacterium*-mediated transient expression of different *TyrA* genes in the leaves of *Nicotiana bethamiana*. Arabidopsis *AtTyrA2*, beet *BvTyrAα* and *GUS* were expressed side-by-side to grass *TyrAs* as controls. Letters indicate significant differences between treatments according to Student's *t*-test ($\alpha = 0.05$; two-sided test for two samples with equal variance). Data presented as the average of $n = 6$ independent plants except for *SbTyrA2* ($n = 5$); error bars = *SD*. Source data for **c–g** are provided within the Source Data file.

Brachypodium[40] and Sorghum[41]. Interestingly, the expression profile of *TyrAs*, in particular *TyrA1* and *TyrA_{nc}*, resembled that of *PTAL*, showing higher expression in stem internodes and roots, and low expression in seeds, flowers, and leaves (Fig. 3b), which correlates with the elevated rate of tyrosine production observed in grass internodes in the $^{13}CO_2$ feeding experiments (Fig. 2). Furthermore, gene co-expression networks in Brachypodium[40], showed that *BdTyrA1* and *BdPTAL* (Bradi3g49250) expression correlate with many other genes of the lignin pathway (Supplemental Table 1). Absolute real-time quantitative PCR (RT-qPCR) comparing *TyrA* expression in young leaves and developing internodes of Brachypodium and Setaria found that *TyrA1* had the highest expression among *TyrA* genes, followed by *TyrA_{nc}* and *TyrA2* (Fig. 3c, d, Supplemental Fig. S4). Importantly, *TyrA1* and *TyrA_{nc}* transcripts were up to 10-times more abundant in young developing internodes than in leaves (Fig. 3c, d). This was not clearly observed for *TyrA2* genes, which showed a fold change comparable to ubiquitin ligase reference genes (~1.5-times; Supplemental Fig. S4). Altogether, these results support that the expression of *TyrA1* and *TyrA_{nc}*, but not *TyrA2*, is strongly induced in developing stems, where active *PTAL* expression and lignin formation take place.

## TyrAnc enzymes, but not TyrA1, exhibit low sensitivity to feedback inhibition by tyrosine

Next, to study the biochemical properties of grass TyrA enzymes, we generated and characterized recombinant purified TyrA proteins from Brachypodium (BdTyrAs) and Sorghum (SbTyrAs), two distantly related grass species[42,43]. Whereas TyrAs from dicot plants are generally $NADP^+$-dependent arogenate dehydrogenases[20,38,44], except for $NADP^+$-prephenate dehydrogenases in the legume family (Fig. 1)[39,45], previous reports on the substrate preference of grass TyrAs are inconclusive[17,19]. Initial biochemical assays to test the substrate and cofactor preference revealed that grass TyrAs are unequivocally most active with arogenate as substrate, rather than prephenate, and $NADP^+$ as cofactor, exhibiting only minor $NAD^+$-arogenate dehydrogenase activity (up to 5% of the main activity; Supplemental Fig. S5). $NAD^+$-prephenate dehydrogenase activity was absent in all cases, ruling out a significant contamination of the enzyme preparations with the TyrA enzyme from *E. coli*[46,47]. Detailed kinetic analyses of the activity with arogenate and $NADP^+$ showed the six TyrAs obey Michaelis-Menten kinetics, with the TyrA_{nc} isoforms having, by a wide margin, the highest turnover number ($k_{cat}$) and lowest Michaelis-Menten constant ($K_m$) (Table 1, Supplemental Fig. S6). Consequently, the catalytic efficiencies ($k_{cat}/K_m$) of BdTyrA_{nc} (~580 $s^{-1}$ $mM^{-1}$) and SbTyrA_{nc} (~337 $s^{-1}$ $mM^{-1}$) were the highest amidst the three isoforms of each species (Table 1).

The activity of most plant TyrA enzymes is competitively inhibited at low concentration of tyrosine, generally in the half maximum inhibition ($IC_{50}$) range of 10 to 50 μM when assayed in vitro[17–20]. Like other plant TyrAs, BdTyrA1 showed an $IC_{50}$ for tyrosine at ~65 μM, and BdTyrA2 had an even lower $IC_{50}$ of ~20 μM (Fig. 3e, Table 1, Supplemental Fig. S6). In contrast, BdTyrA_{nc} exhibited a low sensitivity to

**Table 1 | Kinetic parameters of TyrA enzymes**

| | $K_m$ (mM arogenate) | $k_{cat}$ ($s^{-1}$) | $k_{cat}/K_m$ ($s^{-1} \cdot mM^{-1}$) | $IC_{50}$ (µM tyrosine) | $K_i$ (µM tyrosine) |
|---|---|---|---|---|---|
| BdTyrA1 | $1.41 \pm 0.24$ | $5.2 \pm 0.8$ | $3.6 \pm 0.8$ | $66 \pm 2$ | $46 \pm 3$ |
| SbTyrA1 | $0.57 \pm 0.15$ | $2.9 \pm 0.6$ | $5.3 \pm 0.8$ | $71 \pm 24$ | $53 \pm 10$ |
| BdTyrA2 | $0.45 \pm 0.04$ | $18.5 \pm 0.6$ | $41.2 \pm 1.7$ | $20 \pm 6$ | $8 \pm 2$ |
| SbTyrA2 | $2.13 \pm 0.53$ | $38.5 \pm 8.8$ | $18.1 \pm 9.6$ | $47 \pm 23$ | $36 \pm 13$ |
| BdTyrA$_{nc}$ | $0.13 \pm 0.04$ | $76.5 \pm 5.5$ | $579.6 \pm 24.3$ | $242 \pm 45$ | $64 \pm 25$ |
| SbTyrA$_{nc}$ | $0.22 \pm 0.06$ | $73.5 \pm 18.2$ | $337.5 \pm 19.5$ | $406 \pm 85$ | $137 \pm 44$ |

$k_{cat}/K_m$ was calculated based on $K_m$, $k_{cat}$ and the molecular weight of each recombinant enzyme (including the mass of poly-histidine tag). The half-inhibitory concentration of tyrosine ($IC_{50}$) was calculated at 0.5 mM of arogenate and 1 mM of NADP$^+$ from the data shown in Supplemental Fig. S6. The inhibition constant for tyrosine ($K_i$) was calculated from $K_m$ and $IC_{50}$ values under a competitive inhibition model[19,20]. Data represent average ± $SD$ of $n$ = 4–6 derived from at least two independent experiments conducted on different days using different batches of purified recombinant enzyme. $K_m$ and $k_{cat}$ were calculated from Michaelis–Menten plots shown in Supplemental Fig. S6.

inhibition by tyrosine, with an estimated $IC_{50}$ of ~240 µM. Hence, BdTyrA$_{nc}$ retained >50% of its activity at 200 µM of tyrosine, where BdTyrA1 and BdTyrA2 were fully inactive (Fig. 3e). Despite this marked difference in sensitivity, the inhibition of BdTyrA$_{nc}$ by tyrosine was competitive with arogenate (Supplemental Fig. S7), as reported for other TyrAs[19,20]. Like BdTyrA$_{nc}$, SbTyrA$_{nc}$ also showed low sensitivity to feedback-inhibition, having a high $IC_{50}$ for tyrosine of ~475 µM, whereas SbTyrA1 and SbTyrA2 did not ($IC_{50}$ at 71 and 44 µM, respectively) (Fig. 3f, Table 1).

To investigate if the difference in sensitivity to feedback inhibition impacts the activity of the TyrA isoforms *in planta*, we transiently expressed Brachypodium and Sorghum *TyrA* genes in *Nicotiana benthamiana* through Agrobacterium leaf infiltration (Fig. 3g). As controls, the β-glucuronidase (*GUS*), the tyrosine-inhibited *AtTyrA2* from Arabidopsis[20], and the deregulated *BvTyrAα* from *Beta vulgaris*[44] were also expressed, all under control of the CaMV 35 S promoter ($P_{CaMV35S}$) (Supplemental Fig. S8). Tyrosine content was 2.5 to 3 times higher in *BdTyrA1*, *SbTyrA1*, and *SbTyrA2* expressing leaves compared to the *GUS* control (Fig. 3g). Similar tyrosine levels were observed in the leaves expressing *AtTyrA2*. Overexpression of *BdTyrA2*, which encodes a strongly feedback inhibited enzyme with the lowest $IC_{50}$ among grass TyrAs (Table 1), did not significantly increase tyrosine levels. In contrast, infiltration of the *BdTyrA$_{nc}$* and *SbTyrA$_{nc}$* constructs increased tyrosine content by 8 and 14-times relative to the *GUS* control, respectively, causing an effect similar to the deregulated *BvTyrAα* (Fig. 3g). The *in planta* accumulation of tyrosine correlated better with the sensitivity of the different TyrA enzymes to feedback inhibition ($IC_{50}$), rather than the other kinetical parameters ($k_{cat}$, $K_m$, $k_{cat}/K_m$; Table 1). These results support that, in agreement with their sensitivity to feedback inhibition in vitro, grass TyrA$_{nc}$, but not TyrA1 or TyrA2, can greatly increase tyrosine production when expressed *in planta*.

**Grasses have a feedback insensitive DHS1b enzyme**

Feeding experiments using $^{13}CO_2$ revealed that, beyond high tyrosine production, grass species also synthesize shikimate and phenylalanine at a higher rate than Arabidopsis (Figs. 1 and 2). These findings suggest that the regulation of the upstream shikimate pathway may be different in grass species. To test this hypothesis, we characterized the DHS enzymes from Brachypodium and Sorghum, which catalyze a key regulatory step at the entry point of the shikimate pathway (Fig. 1a)[1,14,24].

The phylogeny of plant DHSs shows that grasses generally have four DHS isoforms (Fig. 4a), which correspond to the Brachypodium loci Bradi1g21330 (namely *BdDHS1a*), Bradi1g60750 (*BdDHS1b*), Bradi3g38670 (*BdDHS2*) and Bradi3g33650 (*BdDHSnc*, from non-

canonical). Whereas *DHS2* and *DHS$_{nc}$* are conserved in other monocots, *DHS1a* and *DHS1b* (which share ~90% of protein sequence identity) are likely derived from a gene duplication event within the grass family. The four Brachypodium *DHS* genes differ in their spatio-temporal expression profile (Fig. 4b). *BdDHS2* is dominant in photosynthetic organs, *BdDHS1a* is expressed across different organs and stages, and *BdDHS$_{nc}$* is mostly expressed in seeds. Notably, *BdDHS1b* expression is induced in the internodes (Fig. 4b), and is co-expressed with *BdPTAL*, *BdTyrA1* and other lignin pathway genes (Supplemental Table 1)[40].

To examine their functional properties, the recombinant DHS enzymes of Brachypodium were produced and characterized in vitro. Though BdDHS$_{nc}$ was also produced, it was not soluble in bacteria and could not be studied. Enzyme assays showed Michaelis–Menten kinetics for phospho*enol*pyruvate, with $K_m$ values in the range of 135 to 200 µM (Table 2, Supplemental Fig. S9), but weak to moderate positive cooperativity for erythrose 4-phosphate, with a $K_{0.5}$–the analogous parameter to $K_m$ in cooperative kinetics–in between 400 and 550 µM (Table 2, Supplemental Fig. S9). $k_{cat}$ values were in the same order of magnitude for the three isoforms (Table 2, Supplemental Fig. S9). Thus, the three Brachypodium DHSs seem to have similar kinetic parameters.

As recent studies have shown that plant DHS enzymes are feedback-inhibited by multiple effector molecules[15,16], we screened the impact of AAAs and 14 related metabolites, including various intermediates of the shikimate pathway and the pathways downstream of AAAs, on Brachypodium DHSs. The effect of these compounds was determined at a concentration of 0.5 mM with two alternative methods: real-time spectrophotometric quantification of phospho*enol*pyruvate consumption[48], and final-point quantification of the reaction product, DAHP, by UHPLC-MS.

Among the three AAAs, phenylalanine did not significantly alter the activity of grass DHSs, which seems to be a common feature in plant DHSs[24]. Tyrosine, which strongly inhibits Arabidopsis DHSs[15], only caused ~25% inhibition in BdDHS1a and ~10% in BdDHS1b and BdDHS2 (Fig. 4C). Conversely, tryptophan strongly inhibited BdDHS2 at an $IC_{50}$ of ~120 µM, but had no effect on BdDHS1a or BdDHS1b (Fig. 4C, D, Table 2). We did not observe any remarkable inhibitory effect caused by intermediates of AAA metabolism or lignin biosynthesis (Supplemental Fig. S10). From the different AAA pathway(s) intermediates tested, we observed that arogenate had the most dramatic effect, causing ~50% inhibition of BdDHS1a, and >75% in BdDHS2 at 0.5 mM (Fig. 4E, Supplemental Fig. S10), with a calculated $IC_{50}$ of ~285 and ~91 µM, respectively (Fig. 4F, Table 2). In contrast, BdDHS1b was not inhibited by arogenate even up to 2.5 mM, when DHS1b activity rather increased (Fig. 4F), likely due to high concentrations of contaminant NaCl present in the arogenate preparation (Supplemental Fig. S11). Under acidic conditions, arogenate is known to undergo a spontaneous dehydration and decarboxylation into phenylalanine[49]. Arogenate incubated with HCl, which is therefore fully converted into phenylalanine, did not inhibit the DHS enzymes, supporting that the inhibitory compound was arogenate instead of other possible contaminants (Supplemental Fig. S11). The characterization of the recombinant Sorghum DHS enzymes confirmed that no strong inhibition was induced by tyrosine or phenylalanine, whereas 0.5 mM of tryptophan caused 40 to 50% inhibition of SbDHS2 and SbDHS$_{nc}$ (Supplemental Fig. S12). Like in Brachypodium DHSs, 0.5 mM arogenate inhibited SbDHS1a and SbDHS2 at ~50% and ~40%, respectively, but had no effect on SbDHS1b.

Determination of the kinetic parameters of BdDHS2 at different concentrations of tryptophan and arogenate showed that both effectors unequivocally decrease $V_{max}$ but had distinct impacts on $K_m$ or $EC_{50}$. For phospho*enol*pyruvate, tryptophan did not significantly changed the $K_m$, which is indicative of non-competitive inhibition, but arogenate increased the $K_m$, indicating a mixed inhibition mechanism (Supplemental Fig. S13). In respect to erythrose 4-phosphate, both

tryptophan and arogenate decreased $K_{0.5}$, suggesting uncompetitive inhibition kinetics (Supplemental Fig. S13). These findings resemble previous studies from bacterial type-II DHS enzymes[50] and support that type-II DHSs, which include plant DHSs, are allosteric enzymes with a complex response to the binding of their substrates and effectors.

DHS effector molecules can have synergistic effects when combined in vitro[50]. To explore this possibility, we tested the impact of different combinations of tryptophan, tyrosine, arogenate and chorismate, at 0.15 mM each, on the activity of Brachypodium DHSs. Although most combinations did not exhibit strong additive effects, some of the combinations, such as tryptophan plus arogenate for

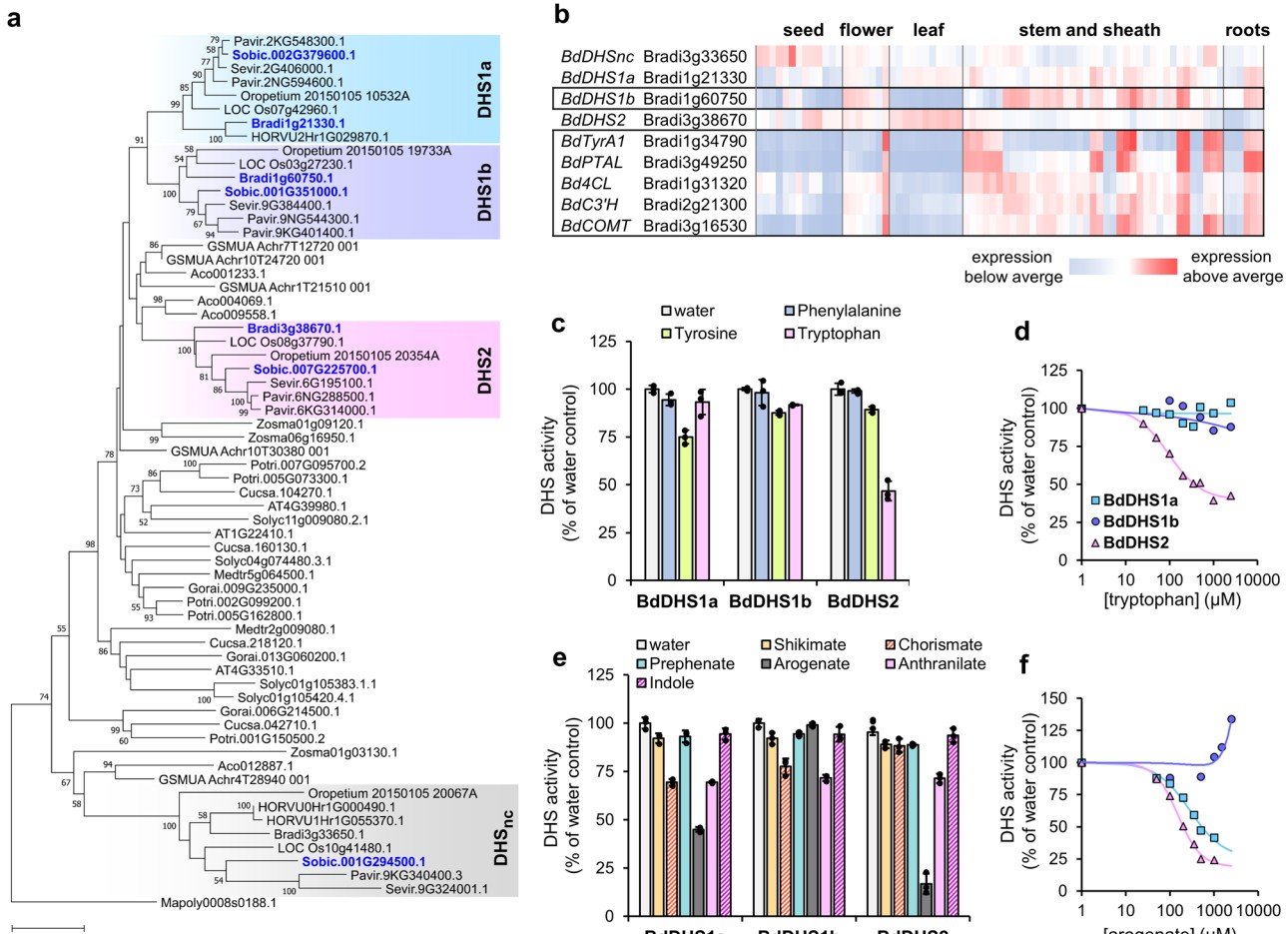

**Fig. 4 | Grasses have a feedback insensitive DHS1b enzyme that is expressed in internodes. a** Phylogeny of plant DHS proteins (outgroup: *Marchantia polymorpha*). Sequences highlighted in blue have been characterized in this study. Bootstrap test values (based on 1000 replications) below 50 have been omitted. Scale bar indicates number of amino acid substitutions per site. Species abbreviations not introduced in Fig. 3a include: Cucsa, *Cucumis sativus*; Gorai, *Gossypium raimondii*, GSMUA, *Musa acuminata*. **b** Expression patterns of Brachypodium *DHS* genes in different organs and developmental stages compared to *BdTyrA1*, *BdPTAL*, and other genes of the lignin pathway. Enzyme abbreviations: 4CL, 4-coumarate:CoA ligase; C3'H, 4-Coumarate 3-hydroxylase; COMT, caffeic acid/5-hydroxyferulic acid O-methyltransferase. **c** Inhibition of recombinant Brachypodium

BdDHS1a, BdDHS1b, and BdDHS2 by 0.5 mM of aromatic amino acids. Data presented as the average of $n = 3$ replicates from three independent experiments; error bars = SD. **d** $IC_{50}$ determination curve of tryptophan inhibition in Brachypodium DHS1a (cyan, squares), DHS1b (purple, circles) and DHS2 (pink, triangles). Data presented as the average of $n = 2$ technical replicates **e** Effect of 0.5 mM of intermediates of AAA pathway(s) on Brachypodium DHSs. Data presented as the average of $n = 3$ replicates from three independent experiments; error bars = SD. **f** $IC_{50}$ determination curve of arogenate inhibition in Brachypodium DHSs (figure legend as in **d**) Data presented as the average of $n = 2$ technical replicates. Source data for panels **c**–**f** are provided within the Source Data File.

## Table 2 | Kinetic parameters of DHS enzymes

| | $K_m$ PEP | $K_{0.5}$ E4P | H Coefficient | $k_{cat}$ | $k_{cat}/K_m$ (PEP) | $k_{cat}/K_{0.5}$ (E4P) | $IC_{50}$ | |
| --- | --- | --- | --- | --- | --- | --- | --- | --- |
| | (mM) | (mM) | (only E4P) | ($s^{-1}$) | ($s^{-1}$ $mM^{-1}$) | ($s^{-1}$ $mM^{-1}$) | (µM Agn) | (µM Trp) |
| BdDHS1a | 0.13 ± 0.02 | 0.55 ± 0.04 | 1.8 ± 0.1 | 17.0 ± 0.2 | 126.0 ± 1.7 | 30.7 ± 0.4 | 285 ± 24 | n.i. |
| BdDHS1b | 0.20 ± 0.07 | 0.40 ± 0.02 | 2.1 ± 0.1 | 11.3 ± 0.1 | 57.2 ± 0.3 | 28.3 ± 0.1 | n.i. | n.i. |
| BdDHS2 | 0.19 ± 0.01 | 0.51 ± 0.10 | 1.5 ± 0.6 | 5.6 ± 0.01 | 29.0 ± 0.1 | 10.8 ± 0.01 | 91 ± 11 | 120 ± 4 |

Original data for the determination of the kinetic parameters $K_m/K_{0.5}$ and $k_{cat}$ are shown in Supplemental Fig. S9. $k_{cat}/K_m$ was calculated as described for TyrA enzymes in the legend of Table 1. $IC_{50}$ for tryptophan and arogenate were determined based on the original data shown in main Fig. 4D, F, respectively. n.i. = not inhibited. All data are means ± SD of n = 4 – 6 derived from at least two independent experiments conducted on different days using different batches of purified recombinant enzyme. PEP phospho*enol*pyruvate, E4P erythrose 4-phosphate.

BdDHS2 and tyrosine plus arogenate for BdDHS1a, showed additive inhibitory effects (Supplemental Fig. S14). None of the combinations tested had a significant impact on BdDHS1b (Supplemental Fig. S14). Hence, although grass DHS1a is inhibited by arogenate and DHS2 by both arogenate and tryptophan, DHS1b seems largely insensitive to feedback inhibition in vitro.

### Co-expression of *BdDHS1b* and *BdTyrA1* synergistically enhances tyrosine production while maintaining high phenylalanine production

To evaluate *in planta* how DHS biochemical regulation may impact the production of AAAs, we expressed *BdDHS1a*, *BdDHS1b*, and *BdDHS2* in *Nicotiana benthamiana* leaves under control of the Arabidopsis *RuBisCO S3B* promoter ($P_{AtRbcS3B}$), which provides 15–20% of the expression level of *CaMV* 35 S promoter[51] (Fig. 5a). Transient expression of *BdDHS1a* and *BdDHS2*, both sensitive to feedback inhibition in vitro, did not significantly alter the content of phenylalanine, tyrosine, tryptophan, nor their common precursor shikimate, compared to the control expressing *tdTomato* (*tdTom*) (Fig. 5a). To the contrary, the expression of *BdDHS1b* triggered the accumulation of 10-times more tyrosine, 3-times more shikimate and tryptophan, and 18-times more phenylalanine, which was the most abundant AAA (~2000 nmol/gFW; Fig. 5a). Quantification of the BdDHS-HA tagged proteins by immunoblotting showed that, despite the marked differences in metabolite levels, the protein levels of these three expressed DHS isoforms were comparable (Supplemental Fig. S15). Taken together, these results support in vitro results showing that grasses possess a naturally deregulated DHS1b that can boost AAA production, mostly phenylalanine, when expressed heterologously *in planta*.

Gene expression data in Brachypodium (Fig. 4b, Supplemental Table 1) indicate that *BdTyrA1* co-expresses with *BdDHS1b* and *BdPTAL* in the internodes, where we detected a high rate of tyrosine production (Fig. 2). Nevertheless, BdTyrA1 is strongly inhibited by tyrosine in vitro (Fig. 3e) and its expression alone in Nicotiana leaves had little impact on tyrosine levels (Fig. 3g). We therefore hypothesized that BdTyrA1 and BdDHS1b may cooperate in the high production of tyrosine and phenylalanine observed in grass tissues. To test this possibility, we co-expressed *in planta* different combinations of *BdTyrAs* and *BdDHSs* and measured the impact of phenylalanine and tyrosine levels. To this end, we took advantage of the Golden Gate modular cloning system[51] and assembled the *BdTyrA* and *BdDHS* expression cassettes into the same vector backbone (Supplemental Fig. S8). To avoid a strong overexpression, *BdTyrA* and *BdDHS* expression was driven by RuBisCO small subunit promoters from Arabidopsis ($P_{AtRbcS3B}$) and tomato ($P_{SlRbcS3A}$), respectively (Supplemental Fig. S8)[50]. Consistent with the results shown in Fig. 3g, the expression of *BdTyrA_nc*, but not *BdTyrA1* or *BdTyrA2*, together with the *tdTom* control led to a strong increase in tyrosine levels (~100-times) (Fig. 5b). Notably, *BdTyrA_nc* expression decreased phenylalanine levels by 4-times compared to the negative control co-expressing *YPet* with *tdTom* (Fig. 5c). Co-expression of *BdTyrA_nc* with *BdDHS1b* showed additive effects and further increased tyrosine to a dramatic ~400-times the *tdTom+YPet* control, while still negatively impacting phenylalanine level (Fig. 5b). Interestingly, co-expression of the feedback-regulated *BdTyrA1* together with *BdDHS1b* boosted tyrosine content to a level close to *BdTyrA_nc*, while still maintaining a high production of phenylalanine (Fig. 5b). This synergistic effect was not observed upon co-expression of *BdDHS1b* with *BdTyrA2*, possibly due to the tight feedback regulation of BdTyrA2 (Table 1). Similarly, no significant

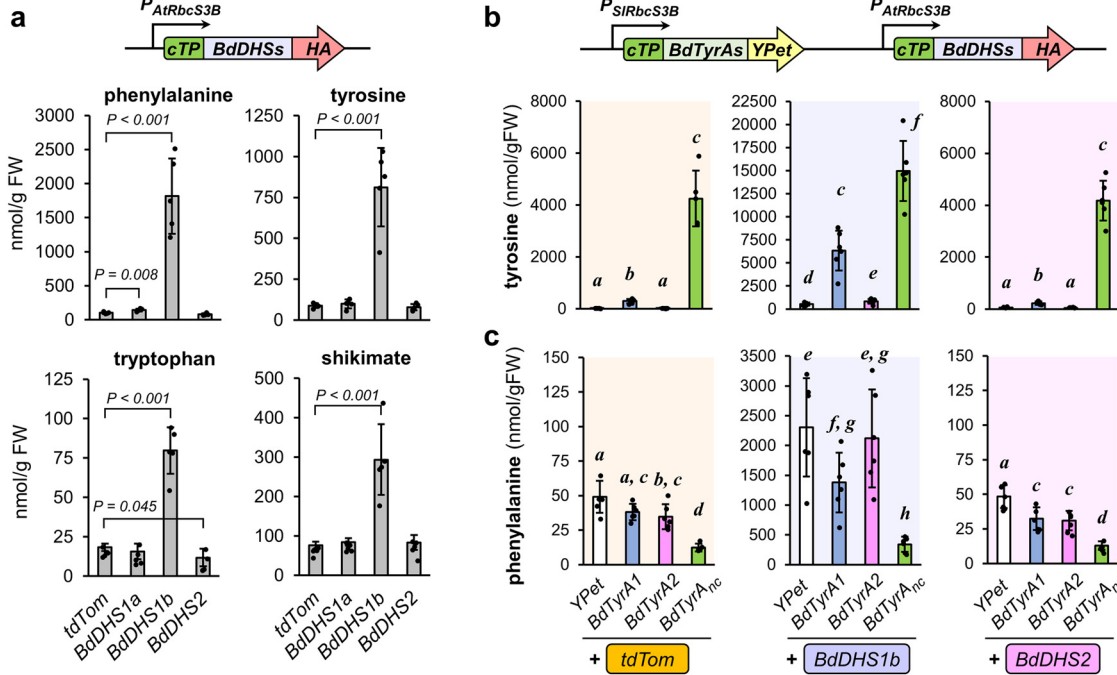

**Fig. 5 | Co-expression of *DHS1b* and *TyrA1* in *Nicotiana benthamiana* have a synergistic impact on tyrosine production while maintaining high phenylalanine accumulation. a** The levels of phenylalanine, tyrosine, tryptophan, and their common intermediate shikimate, three days after the transient expression of Brachypodium *DHS1a*, *DHS1b* and *DHS2* in the leaves of *Nicotiana benthamiana* under control of a RuBisCO promoter. Note that different scales have been used for the individual panels. Data presented as the average of $n = 5$ individual plants; error bars = SD. P values for statistically significant differences ($\alpha = 0.05$) with respect to the *tdTomato (tdTom)* negative control are based on Student's *t*-test (two-sided test

for two samples with equal variance). **b** Impact of the co-expression of *BdDHS1b* or *BdDHS2* with *BdTyrA1*, *BdTyrA2*, or *BdTyrA_nc* on tyrosine and **c** phenylalanine accumulation in *Nicotiana benthamiana* leaves. Note that different scales have been used for the individual panels. Data presented as the average of $n = 6$ individual plants except $n = 5$ for *tdTom+BdTyrAnc* treatment; error bars = SD. Letters indicate significant differences between treatments according to Student's *t*-test ($\alpha = 0.05$; two-sided test for two samples with equal variance). Source data are provided within the Source Data File.

additive effects were observed upon co-expression of the *BdTyrAs* with the feedback-regulated *BdDHS2* (Fig. 5b, c). These results show that simultaneous expression of deregulated BdDHS1b with the feedback-regulated BdTyrA1 can render high levels of both tyrosine and phenylalanine *in planta*.

## Discussion

The emergence of phenylpropanoid metabolism is a key adaptation during the transition of plants from water to land, by conferring plants with enhanced mechanical strength, and protection against UV radiation and desiccation. Phenylpropanoids are remarkably diverse across the plant phylogeny but synthesized exclusively from phenylalanine and by the PAL enzyme in almost all plant groups[5,52,53]. Grasses constitute an exception, as they use the PTAL reaction to synthesize phenylpropanoids from tyrosine, introducing a shortcut in the canonical phenylpropanoid pathway (Fig. 1a)[5,54]. In this study, we used this unique grass feature to investigate how enzyme evolution re-shapes metabolic flows and regulation at the interface of primary metabolism and the downstream natural product pathways.

Labeling experiments using $^{13}CO_2$ provided well-grounded evidence that supports high tyrosine production in grasses, especially in the internodes (Figs. 1 and 2). Consistently, the expression of *TyrA* genes was induced in lignifying tissues of grasses (Fig. 3), unlike in dicots where *ADT* but not *TyrA* genes are co-expressed with lignin pathway genes[29,55,56]. One of the grass TyrAs, TyrA_{nc}, also showed low sensitivity to inhibition by tyrosine in vitro, and its expression in *Nicotiana benthamiana* boosted tyrosine levels (Figs. 3f and 5b), though at the expense of phenylalanine production (Fig. 5c). These findings clearly support that, unlike most plant TyrAs[18–20], grass TyrAnc has low sensitivity to feedback inhibition. Partially feedback insensitive TyrA enzymes have been previously described in Caryophyllales, with $IC_{50}$ values between ~400 and 700 μM of tyrosine[44,57], and in legumes, where the prephenate dehydrogenase TyrA_{p} is completely feedback insensitive[39,45]. However, phylogenetic evidence shows that grass TyrA_{nc} are not related to Caryophyllales TyrA_{α} or legumes TyrA_{p} (Fig. 3a), indicating that feedback de-regulated TyrAs likely evolved independently in these three plant groups.

Despite their highly active tyrosine biosynthesis, grasses still maintain a high rate of phenylalanine production, particularly in stems (Figs. 1 and 2). Detailed characterization of grass DHS isoforms uncovered that DHS1b, unlike DHS1a and DHS2, is insensitive to feedback inhibition (Fig. 4). The expression of *DHS1b*, but not *DHS1a* or *DHS2*, in Nicotiana leaves also boosted the level of shikimate, tryptophan, tyrosine, and especially phenylalanine (Fig. 5a). Although feedback-insensitive mutant DHSs have been introduced in plants to increase the production of AAAs and/or phenylpropanoids[21,32,58–60], DHS1b constitutes, to our knowledge, the first report of a naturally occurring deregulated plant DHS.

Previous studies on biochemical characterization of DHS activity from plant extracts—most of them dicots—reported varying observations about the sensitivity of DHS to feedback regulation[24], likely due to the presence of multiple DHS isoforms in plants[24,61]. In monocots, a single study found that DHS activity in crude extracts from 9-day-old maize plantlets is inhibited by tryptophan, but not by phenylalanine or tyrosine[62], consistent with a predominant role of the tryptophan-inhibited DHS2 in green tissues (Fig. 4). Based on the differences in feedback regulation and expression patterns of individual grass DHS isoforms (Fig. 4), DHS1a and DHS2 may display more general or house-keeping roles, whereas DHS1b may have organ-specific functions (e.g., internodes) that demand high AAA production. Recent studies in Arabidopsis also showed a complex isoform-dependent feedback regulation where different DHS isoforms are expressed and regulated differently[15,16]. Moreover, Arabidopsis DHSs were inhibited by chorismate, caffeic acid, tyrosine, and its derived metabolites 4-hydroxyphenylpyruvate and homogentisic

acid[38], which all had little effect on grass DHS enzymes (Fig. 4; Supplemental Figs. S10, S12). In the case of tyrosine and its derived compounds 4-hydroxyphenylpyruvate and homogentisic acid, the insensitivity of grass DHSs might be linked to the high tyrosine levels present in grass tissues (Figs. 1 and 2). Therefore, it seems that the feedback regulation of plant DHSs is not only isoform-dependent, but also species-dependent. These findings indicate that both biochemical and transcriptional regulatory mechanisms targeting DHSs give plant species a precise yet adaptable tool to modulate AAA production.

Although the feedback insensitive TyrAnc is likely contributing to the high rate of tyrosine production in grass internodes (Fig. 2), it is somewhat incongruent that *TyrA1*, whose expression is strongly induced in the internodes (Fig. 3c, d), encodes a tyrosine-inhibited enzyme (Table 1). Interestingly, the combinatorial expression of the feedback-inhibited TyrA1 with the deregulated DHS1b led to high production of both tyrosine and phenylalanine *in planta* (Fig. 5). This synergistic effect might be a consequence of "pushing" the carbon flow into the shikimate pathway, causing an accumulation of arogenate that would alleviate the competitive feedback inhibition of TyrA1 by tyrosine (Supplemental Fig. S7)[19,20] and support high rates of both tyrosine and phenylalanine production. The relatively high $K_m$ values of grass TyrA1s for arogenate (Table 1) may reflect their specialization for working at the high substrate levels provided by DHS1b, while TyrA_{nc} is possibly better suited for producing tyrosine at low arogenate concentrations thanks to its lower $K_m$ value (Table 1).

Overall, the current findings highlight that the interplay between feedback-regulated (TyrA1) and deregulated (DHS1b, TyrA_{nc}) enzymes at the entry and exit steps of AAA biosynthesis can maintain the high production of both tyrosine and phenylalanine. This fine-tuning of the upstream AAA pathway likely supports the unique dual lignin pathway found in grasses. Future studies of these key enzymes from different monocot species will address the evolutionary history of the coordinated regulation of the grass AAA and lignin pathways. The fundamental knowledge also provides useful genetic tools for the rationale engineering of plant primary metabolism to support the production of aromatic products.

## Methods

### Plant materials and growth conditions

The following grass cultivars were used in this study: *Brachypodium distachyon* 21-3, *Sorghum bicolor* RTx430, and *Setaria viridis* A10.1. Sorghum leaves used for cloning were a courtesy of the Wisconsin Crop Innovation Center (Middleton, Wisconsin).

Arabidopsis, Brachypodium and Setaria plants used for $^{13}CO_2$ feeding and RT-qPCR analysis were kept in a growth chamber at 22 °C, 12h-photoperiod under ~100 μE of light intensity, 60% humidity, and watered with a 1:10 dilution of Hoagland's solution.

*Nicotiana benthamiana* plants used for transient expression experiments were grown at 22 °C in a 12-hours photoperiod under ~200 μE of light intensity, 60% humidity, and watered with a 12:4:8 (N:P:K) plant nutritive solution (Miracle-Gro) at a 1:1000 dilution.

### Gene expression analysis and RT-qPCR

Spatio-temporal gene expression data *Brachypodium distachyon* and *Sorghum bicolor* were retrieved from PlaNet[40] and the MOROKOSHI database[41], respectively.

Total RNA was isolated from young leaves and developing internodes of 1.5-months old Brachypodium and Setaria plants using RNeasy Plant Mini Kit (Qiagen), following manufacturer's instructions. RNA was treated with RQ1 RNase-free DNase (Promega) and reverse transcribed with M-MLV Reverse Transcriptase (Promega) using random hexamer primers. Quantitative PCR analysis was carried out in a Stratagene Mx3000P (Agilent Technologies) thermocycler using GoTaq qPCR Master Mix (Promega). Ct values were determined using

LinRegPCR[63] version 2018.0. Primers used are listed on Supplemental Table 2. Ct values were converted into mass of template by using a calibration curve made of the corresponding RT-qPCR amplicon, cloned into the EcoRV site of pML94 vector using conventional blunt-end ligation protocols. Ubiquitin ligase genes of Brachypodium and Setaria were chosen as reference genes based on previous publications[64].

## Enzyme assays

TyrA assays were conducted in a plate reader at 37 °C (Tecan Infinite M Plex, Tecan) using half-area plates (Greiner Bio-One) by tracking the conversion of NAD(P)$^+$ into NAD(P)H as the increment of absorbance at 340 nm. TyrA reactions consisted of a final volume of 50 µL of 50 mM HEPES buffer pH 7.5, 50 mM KCl, 1 mM NADP$^+$ (NAD$^+$), and the enzyme (variable concentration, see details below). For $IC_{50}$ assays, tyrosine was included into the reaction mixture pipetted from 10X-stocks adjusted to pH-10 with NaOH, as tyrosine solubility is low at neutral pH. Enzyme concentration was adjusted using TyrA desalting buffer (see "Protein expression and purification") supplemented with bovine serum albumin (BSA, protease-free powder purified by heat shock process; Fisher bioreagents), to ensure at least 3 min of linear reaction. For arogenate-NADP$^+$ activity, the mass of enzyme was adjusted to 10 to 200 ng per reaction, depending on the specific activity of the TyrA isoform being tested. For assays using NAD$^+$ and/or prephenate, the enzyme mass per reaction was upscaled to 200–1000 ng to increase sensitivity. The reaction mixtures with the enzyme and without substrate (arogenate or prephenate) were incubated at 37 °C for 3 minutes, upon the addition of the substrate at the following final concentration, depending on the experiment: for determination of the enzyme substrate, 1 mM of prephenate or arogenate; for $K_m$ and $k_{cat}$ determination, variable concentrations up to 2.5 mM µM of arogenate; for $IC_{50}$ determination, 0.5 mM of arogenate.

Except when specified (Supplemental Fig. S9) DHS activity was measured using a real-time method by tracking the consumption of phosphoenolpyruvate at 232 nm[48] at 37 °C in a plate reader (Tecan Infinite M Plex, Tecan) in half-area UV-transparent 96-well plates (UV-Star®, Greiner Bio-One). DHS reaction consisted of a final volume of 50 µL of 25 mM HEPES buffer pH 7.5, 2 mM MgCl$_2$, 3 mM dithiothreitol (DTT), the enzyme (variable mass, see details below), the effector (if tested) and the substrates (phosphoenolpyruvate and erythrose 4-phosphate). All DHS effectors tested were included in the initial reaction mixtures at a concentration of 0.5 mM, except for $IC_{50}$ determination of arogenate and tryptophan, in which variable concentrations were used. To ensure at least 10 minutes of linear reaction, enzyme mass per reaction was carefully adjusted between 100 and 300 ng (depending on the specific activity of each specific isoform) using DHS storage buffer (see "Protein expression and purification") supplemented with BSA. The reaction mixtures, having the enzyme and all the other components except substrates, were incubated for 5 minutes at room temperature to allow the DTT-mediated activation of DHS. After this, phosphoenolpyruvate (variable concentration, see below) was mixed into the reaction, and a second incubation step of 5 minutes at 37 °C was performed. The enzymatic reaction was started with the addition of erythrose 4-phosphate (variable concentration, see below). The following concentrations of substrates were used depending on the specific experiment: for testing potential feedback inhibitors and determination of $IC_{50}$, 1.5 mM phosphoenolpyruvate and 2 mM erythrose 4-phosphate; for calculating $K_m$ and $k_{cat}$ for phosphoenolpyruvate, fixed erythrose 4-phosphate at 2 mM and variable concentrations of phosphoenolpyruvate up to 2 mM; for calculating $K_{0.5}$ and $k_{cat}$ for erythrose 4-phosphate, fixed phosphoenolpyruvate at 1.5 mM and variable erythrose 4-phosphate concentrations up to 3 mM.

For DHS effector molecules overlapping with phosphoenolpyruvate absorbance in the UV range, DHS activity was contrasted by a final-point quantification of the reaction product DAHP by UHPLC-MS. The DHS assay for UHPLC-MS quantification was set up using the same settings as described in the previous paragraph for the UV-based DHS assay, which guarantee >10 minutes of reaction linearity. After 10 minutes incubation, 20 µL of the reactions (out of a total volume of 50 µL) were mixed into 80 µL of methanol, vortexed, spun down at 20,000 g for 5 minutes and transferred to vials for injection. Analysis of DAHP by UHPLC-MS was conducted using the same chromatographic settings as described for the UHPLC-MS analysis of soluble metabolites and compared with an authentic DAHP standard (Sta. Cruz bio-technology, cat. no. sc-216432).

Kinetic parameters of both TyrAs and DHSs were determined in MS-Excel using Solver add-in function. Arogenate was prepared by enzymatic conversion from prephenate (Prephenate Barium salt, Sigma-Aldrich), as previously described[65].

## Transient expression experiments in Nicotiana benthamiana

Agrobacterium tumefaciens strain GV3101 transformed with the plant expression constructs were grown at 28 °C for 24 to 36 hours in 10 mL of LB liquid media containing the corresponding antibiotics. The saturated cultures were spun down at 3,000 g for 5 minutes at room temperature and washed twice with 3 mL of induction media (IM; 10 mM MES [2-(N-morpholino)ethanesulfonic acid] buffer pH 5.6, 0.5% glucose, 2 mM NaH$_2$PO$_4$, 20 mM NH$_4$Cl, 1 mM MgSO$_4$, 2 mM KCl, 0.1 mM CaCl$_2$, 0.01 mM FeSO$_4$, and 0.2 mM acetosyringone). After washing, bacteria cultures were incubated in IM for 2 to 3 hours at room temperature in the dark, pelleted at 3000 g for 5 minutes and resuspended into 3 mL of 10 mM MES buffer pH 5.6 with 0.2 mM acetosyringone. OD$_{600nm}$ was adjusted to a final density of 0.25 units for pAGM4673::TyrA (Fig. 3g) and pICH47822::DHS (Fig. 5a) infiltration, or 0.5 units for pAGM4673::TyrA-DHS co-expression constructs (Fig. 5b, c) using 10 mM MES buffer pH 5.6 with 0.2 mM acetosyringone. For infiltration pICH47822::DHS constructs, the Agrobacterium suspensions were adjusted to OD$_{600nm}$ = 0.5 and mixed with an equal volume of a suspension of an Agrobacterium clone transformed with pICH4780::p19 under control of Arabidopsis Ubiquitin ligase promoter adjusted to OD$_{600nm}$ = 0.5, resulting in a final mixture of 0.25 OD$_{600nm}$ units for each construct. The inclusion of p19 gene silencing suppressor was found to be particularly critical to express grass DHS genes. Nicotiana benthamiana plants of around 4-weeks-old were infiltrated close to the end of the light period into four different spots per plant, distributed into two leaves at two infiltrations per leaf, with each individual spot corresponding with a different construct/treatment. In total, each construct was infiltrated as 5 or 6 independent replicates into different plants following a randomized pattern. Samples consisting of the infiltrated leaf limbs, without the main veins, were harvested at -72 hours post infiltration and subjected to HPLC or UHPLC-MS analysis (see details below).

## $^{13}CO_2$ feeding

Brachypodium, Setaria and Arabidopsis plants were grown in 2.5 × 2.5 inches pots and randomly distributed into a plexiglass labeling chamber of approximately 32 liters of total volume. The artificial air mixture containing 79% N$_2$, 21% O$_2$ and 0.040% (400 ppm) of $^{13}CO_2$ was pumped at a normal flow rate of 2 liters per minute. The air flow was connected 15 minutes before the beginning of the light period. For sampling, the air flow was interrupted, and the plant samples (entire plants for the experiment represented in Fig. 1; fully expanded leaves and stem tissue for Fig. 2) harvested and frozen immediately into liquid nitrogen. Feeding was resumed by reconnecting the air flow at 10 liters per minute with no $^{13}CO_2$ included in the mixture to quickly purge atmospheric $^{12}CO_2$. After 5 minutes, the flow rate was resumed at 2 liters per minute, -400 ppm $^{13}CO_2$, and kept constant until the next sampling time. Light intensity and temperature during the experiment were -100 µE and 22 °C, respectively.

## UHPLC-MS/MS analysis of metabolites

Around 30–40 mg of pulverized frozen plant tissue were resuspended into 400 μL of chloroform:methanol (1:2) for ~1 hour with regular vortexing, followed by centrifugation at 20,000 g for 5 minutes at room temperature. The supernatant was transferred to a fresh tube, mixed with 125 μL of chloroform, 300 μL of water, and spun down at 15,000 g for 5 minutes for phase separation. The upper, aqueous phase was recovered and dried down for 4 hours to overnight in a speed-vac at 40 °C. The dried pellets were resuspended into 100 μL of methanol 80%, spun down at 20,000 g for 5 minutes, and the supernatant transferred to vials for injection. All reagents used for the extraction were UHPLC-MS grade.

Aromatic amino acids and shikimate were detected using a Vanquish Horizon Binary UHPLC (Thermo Scientific) coupled to a Q Exactive mass spectrometer (Thermo Scientific). One microliter of the sample was analyzed using an InfinityLab Poroshell 120 HILIC-Z column (150 × 2.1 mm, 2.7-μm particle size; Agilent) in a gradient of 5 mM ammonium acetate/0.2% acetic acid buffer in water (solvent A) and 5 mM ammonium acetate/0.2% acetic acid buffer in 95% acetonitrile (solvent B) at a flow rate of 0.45 mL/min and column temperature of 40 °C. The phase B gradient was: 0–2 minutes, 94%; 2–9 minutes, 94–88%; 9–19 minutes, 88–71%; 19–20 minutes, 71–20%; 20–21.5 minutes, 20%; 21.5–22 minutes, 20–94%; 22–25 minutes, 94%. All chemicals used to prepare the mobile phases were LC-MS grade. Full MS spectra were recorded between 2 and 19 min using full scan in negative mode, under the following parameters: sheath gas flow rate, 55; auxiliary gas flow rate, 20; sweep gas flow rate, 2; spray voltage, 3 kV; capillary temperature, 400 °C S-lens RF level, 50; resolution, 70,000; AGC target $3 \times 10^6$, maximum scan time 100 ms; scan range 70–1050 m/z. Spectral data were integrated manually using Xcalibur 3.0. For $^{13}C$ labeled plant samples, $^{13}C$-isotopologues were detected based on a mass increase of 1.00335 atomic mass units for each $^{13}C$ atom. Compound abundance was calculated based on high purity standards: Amino Acid Standard H for tyrosine and phenylalanine (Thermo Scientific, cat. no. PI20088), and shikimic acid ≥99% (Millipore Sigma, cat. no. S5375).

## Determination of tyrosine content by HPLC in *Nicotiana benthamiana* extracts

The infiltrated leaf areas, excluding the midrib and major veins, were harvested at 72 h after infiltration and frozen immediately into liquid nitrogen. The extraction protocol for analyzing tyrosine by HPLC was based on previous studies[66], and consisted of extracting 15 to 25 mg of pulverized frozen plant tissue into 400 μL of 0.5% 2-amino-2-methyl-1-propanol buffer pH 10.0 in 75% ethanol for 1 hour with regular vortexing, followed by centrifugation at 20,000 g for 5 minutes. 300 μL of the supernatant were then transferred to a fresh tube, dried down in a SpeedVac at ~40 °C. The pellets were resuspended into 100 μL of water, spun down at 20,000 g for 10 minutes, and transferred to HPLC vials for injection. HPLC analysis was conducted in the model Infinity 1260 (Agilent, Santa Clara, CA) equipped with a Water's Atlantis T3 C18 column (3 μ, 2.1 × 150 mm) using mobile phases of A (water with 0.1% formic acid) and B (acetonitrile with 0.1% formic acid) in a 20 minutes gradient of the mobile phase B: 0 to 5 miutesn, 1% isocratic; 5 to 10 minutes, linear increase from 1% to 76%; 10 to 12 minutes, linear decrease from 76% to 1%; 12 to 20 minutes, 1% isocratic. Tyrosine peak was detected at the retention time ~3.5 minutes using fluorescence detection mode (excitation wavelength 274 nm, emission wavelength 303 nm) and quantified with an authentic tyrosine standard (Alfa Aesar, catalog number AAA1114118).

## TyrA sequence identification and phylogenetic analysis

TyrA and DHS protein sequences were downloaded from Phytozome[67] v13 using pBLAST search in the following genomes (species abbreviations between parenthesis): *Amaranthus hypochondriacus* v2.1 (AH),

*Ananas comosus* v3 (Aco), *Arabidopsis thaliana* TAIR10 (At), *Beta vulgaris* EL10_1.0 (EL), *Chlamydomonas reinhardtii* v5.6 (Cre), *Cucumis sativus* v1.0 (Cucsa), *Brachypodium distachyon* v3.2 (Bd), *Gossypium raimondii* v2.1 (Gorai), *Hordeum vulgare Morex* v3 (HORVU), *Marchantia polymorpha* v3.1 (Mapoly), *Medicago truncatula* Mt4.0v1 (Medtr), *Musa acuminata* v1 (GSMUA), *Oropetium thomaeum* v1.0 (Oropetium), *Oryza sativa* v7.0 (LOC_Os), *Panicum virgatum* v5.1 (Pavir), *Phaseolus vulgaris* v2.1 (Phvul), *Populus trichocarpa* v4.1 (Potri), *Selaginella moellendorffii* v1.0 (Selmo), *Setaria viridis* v2.1 (Sevir), *Solanum lycopersicum* ITAG4.0 (Solyc), *Spinacia oleracea* Spov3 (Spov), *Sorghum bicolor* v3.1.1 (Sb), *Zea mays* RefGen_V4 (Zm), *Zostera marina* v3.1 (Zosma). Protein sequences without the putative plastid transit peptide were aligned using MUSCLE in MEGA-11[68]. Phylogenies were reconstructed in MEGA-11 using the Neighbor-Joining method and a site coverage cutoff was set at 90%. The tree is drawn to scale, with branch lengths in the same units as those of the evolutionary distances used to infer the phylogenetic tree. The evolutionary distances were computed using the Poisson correction method, and are in the units of the number of amino acid substitutions per site. Bootstrap values were calculated based on 1000 replications.

## Cloning of *TyrA* and *DHS* genes into pET28a

Plant total RNA used for cloning was extracted from young leaf tissue using the CTAB/LiCl method[69] with modifications[70]. cDNA was synthesized with SuperScript IV VILO Master Mix (Thermo Scientific) following manufacturer's instructions.

All genes were cloned without the predicted plastid transit peptide (TargetP v2.0 server, DTU Health Tech) using specific primers listed in Supplemental Table 2. *TyrA1* and *TyrA2* genes were directly cloned from genomic DNA, as these genes lack introns. Grass *TyrA_{nc}* genes were cloned from cDNA of the corresponding species. *DHS* genes from Brachypodium were cloned from cDNA. *SbDHS1a* and *SbDHS2* were cloned from *Sorghum bicolor* cDNA. *SbDHS1b* and *SbDHSnc* were gene synthesized into pET28a vector (GeneArt, Thermo-Fisher). All cloning PCRs were conducted using high fidelity DNA polymerase (PrimeSTAR Max DNA polymerase, Takara Bio). PCR amplicons were purified from gel using QIAquick gel extraction kit (QIAGEN) and cloned into the pET28a vector between *Nde*I and *Bam*HI sites by In-Fusion cloning (Clontech). All cloned genes were confirmed by Sanger sequencing.

## Plant expression constructs

For transient expression of *TyrA* genes of Brachypodium in Arabidopsis protoplasts, the full-length CDSs, without stop codon, were amplified by PCR from cDNA (*BdTyrA_{nc}*) or genomic DNA (*BdTryA1* and *BdTyrA2*, which lack introns) using corresponding gene-specific primers (Supplemental Table 2). cDNA was prepared as described for pET28a constructs. The PCR fragments were purified from gel and inserted into the vector backbone pML94 at *Kpn*I and *Not*I sites, using the In-Fusion cloning (Clontech). The constructs were confirmed by restriction digestion and Sanger sequencing.

For *TyrA* expression in *Nicotiana benthamiana* under control of CaMV 35 S promoter, the *TyrA* genes were amplified from pET28a constructs and assembled into a modified version of the binary vector pAGM4673 (Addgene plasmid #48014, courtesy of Sylvestre Marillonnet[71], Supplemental Fig. S8) using *Bsa*I sites introduced downstream of the CaMV 35 S promoter. The plastid transit peptide from the enzyme 3-*enol*pyruvylshikimate 3-phosphate synthase from *Petunia × hybrida* was used to target the TyrA proteins into the plastid[72].

For the simultaneous expression of Brachypodium *TyrA* and *DHS* genes in *Nicotiana benthamiana*, the genes were first cloned into the level 0 backbone pAGM1287 (Addgene plasmid #47996, courtesy of Sylvestre Marillonnet[71]) by In-Fusion cloning (Clontech). The level 0 modules were assembled into the level 1 binary vector pICH47831 for

*TyrAs*, or pICH47822 for *DHSs* (Addgene plasmids #48009 and #48010, courtesy of Sylvestre Marillonnet[71] as illustrated in Supplemental Fig. S8), using the MoClo Plant Parts Kit (Addgene Kit # 1000000047, courtesy of Nicola Patron[50]). The level 1 modules were then transferred into the level 2 binary backbone pAGM4673 (Supplemental Fig. S8). All constructs were checked by restriction digestion and Sanger sequencing prior to being transformed into *Agrobacterium tumefaciens* GV3101. All primers used are listed in Supplemental Table 2.

### Protein expression and purification
Recombinant proteins were produced using the *E. coli* strains Rosetta-2 (DE3) (Millipore Sigma) for TyrA$_{nc}$, ArcticExpress (Agilent) for TyrA1s and TyrA2s, and KRX (Promega) for DHSs. In all cases, starter cultures were grown overnight at 37 °C, 200 rpm in 10 mL terrific broth (TB) medium containing the corresponding pET28a antibiotic (50 μg/mL kanamycin) and 0.1% glucose. Next day, flasks containing 200 or 400 mL of TB medium with 50 μg/ml kanamycin and without glucose were inoculated with a 1:100 dilution of the starter cultures, and kept at 37 °C, 200 rpm, until OD$_{600\,nm}$ reached ~0.5–0.6. For TyrA$_{nc}$ production in Rosetta-2, the cultures were cooled down to room temperature for ~15 minutes, induced with 0.5 mM of isopropyl β-D-1-thiogalactopyranoside (IPTG), and kept at 22 °C, 200 rpm, for 8 to 10 hours. For the production of DHS proteins in KRX, the cultures were cooled down to room temperature for ~15 minutes, induced with 0.5 mM IPTG and 0.1% rhamnose, and kept at 22 °C, 200 rpm, for 16–20 hours. For TyrA1 and TyrA2, ArcticExpress cultures were cooled down in a mixture of water and ice for ~10 minutes, induced with 0.5 mM IPTG and kept at 15 °C, 200 rpm, for 16 to 20 hours. All cultures were pelleted at 5000 *g* for 10 minutes and stored at −80 °C until purification.

Frozen bacterial pellets were thawed on ice and resuspended into 2 to 4 mL of LEW buffer (Lysis-Equilibration-Washing buffer; 50 mM sodium phosphate buffer pH 8.3, 300 mM NaCl 300 mM and 10% v/v glycerol) supplemented with 1 mM phenylmethylsulfonyl fluoride (PMSF) and 1 mg/mL lysozyme, and sonicated on ice for 5 minutes in 30 seconds cycles. Cell lysate was centrifuged at 15,000 *g*, 4 °C, for 15 minutes. The supernatant was recovered, mixed with 100 uL of PureProteome Nickel Magnetic Beads (Millipore) previously washed with LEW buffer, and kept in the cold under gentle shaking for 30 minutes for binding. After that, the magnetic beads were washed twice with 1 mL of LEW buffer. Proteins were eluted with LEW buffer with 250 mM imidazole into four fractions of 100 μL each. The fraction(s) with the highest protein concentration (usually two) were combined and exchanged into the corresponding storage buffer using Sephadex G-50 resin (GE Healthcare): for TyrA proteins, 50 mM 4-(2-hydroxyethyl)-1-piperazineethanesulfonic acid (HEPES) buffer pH 7.5, 50 mM KCl, 10% glycerol, and 1 mM DTT; for DHS proteins, 50 mM HEPES buffer pH 7.5, 300 mM NaCl, and 0.2% Triton X-100. In the case of DHS proteins, keeping NaCl concentration at >150 mM in the storage buffer was found to be critical to prevent protein precipitation. After buffer-exchange, proteins were frozen immediately in liquid nitrogen and stored at −80 °C. Concentration of total protein was determined using Bio-Rad Protein Assay Dye Reagent Concentrate (Bio-Rad). Purity level of the recombinant enzymes was determined in ImageJ (v1.52a) upon staining of SDS-PAGE gel with Coomassie Brilliant Blue R-250. Enzymatic assays were carried out within no longer than 2 weeks of protein storage at −80 °C, although we found many TyrAs and DHSs to be stable for longer periods (few months) under these conditions.

### Plastid targeting assay in *Arabidopsis* protoplasts
Localization studies for BdTyrA1, BdTyrA2, and BdTyrA$_{nc}$ were performed in Arabidopsis protoplasts using C-terminal fusion to EGFP. Plasmid DNA was isolated from *E. coli* cell cultures with the PureYield™ Plasmid Maxiprep System (Promega). Protoplasts were isolated from

two-weeks-old *Arabidopsis thaliana* leaves, transfected with plasmid DNA and incubated for 16 h to allow for protein expression and maturation. Samples were analyzed by laser scanning confocal microscopy using a Zeiss LSM 780 ELYRA PS1 (Newcomb Imaging Center, Department of Botany, UW-Madison). The light path included a 488 nm and a 561 nm laser and a 488/561 dichroic mirror. Fluorescence was detected in two tracks in the range of 578–696 nm and 493–574 nm to record chlorophyll autofluorescence and EGFP signal, respectively. All images were captured with a LDC-Apochromat 40×/1.1 W Korr M27 objective. Images were processed using Zen software (Zeiss).

### Extraction of plant proteins and western blot
Total proteins from *Nicotiana benthamiana* samples were extracted from ~10 mg of pulverized frozen tissue into 75 μL of 1X denaturing protein sample buffer (60 mM Tris [tris(hydroxymethyl)aminomethane] buffer pH 6.8, 2% sodium dodecyl sulfate, 10% glycerol, 3% β-mercaptoethanol, and 0.01% bromophenol blue) by vigorous vortexing for 30 s and boiled immediately at 95 °C for 7 minutes. Tubes were centrifuged at 15,000 *g* for 5 minutes and 5 μL of the supernatant were applied per lane to the SDS-PAGE gel. Proteins were transferred to a PVDF membrane and blocked for 1 hour in 5% skimmed milk in Tris saline buffer (TBS) with 0.05% Tween-20 before incubation with the corresponding antibodies. HA tagged fusion proteins were detected using an anti-HA tag monoclonal antibody conjugated to horseradish peroxidase (HRP) at a 1:1,000 dilution (HA-Probe HRP conjugated mouse monoclonal antibody clone F-7, Sta. Cruz Biotechnology, cat. no. SC-7392). Antibody dilutions were prepared in TBS buffer with 0.05% Tween-20 and 0.5% BSA. Immunoblot signal was quantified in non-saturating conditions using ImageJ (version 1.52a) and pure recombinant BdDHS1b-3xHA as standard, which was mixed with total protein extracts of not-infiltrated *Nicotiana* leaves to ensure homogenous transfer for all lanes. Independent western blot membranes were exposed in parallel to ensure quantitative results. For details about the generation of the recombinant protein standards, see section "Protein expression and purification" in materials and methods.

### Statistics and reproducibility
All experiments shown in the manuscript were conducted at least twice to confirm the reproducibility of the findings. Statistical comparisons between samples (Student's *t*-test) were performed using Microsoft Office Excel. No data were excluded from any of the analyses shown, except for one datapoint in Figs. 3g and 5b, c, due to technical issues while extracting or analyzing the plant samples (see corresponding figure legends and Source Data Set for more information). Original data for Figs. 1–5 are provided within the Source Data File associated with this article. No statistical method was used to predetermine sample size. All the experiments conducted with plants were randomized. The Investigators were not blinded to allocation during experiments and outcome assessment.

### Reporting summary
Further information on research design is available in the Nature Portfolio Reporting Summary linked to this article.

## Data availability
The original data supporting the findings of this study are provided with this paper as a Source Data File. All published data are available without restrictions. The sequence of the genes studied in this work are publicly available at Phytozome v13, and their IDs clearly indicated at the first time introduced. Source data are provided with this paper.

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

## Acknowledgements

We are thankful to Dr. Ray Collier (Wisconsin Crop Innovation Center; Middleton, Wisconsin) for his assistance in designing and assembling pAGM4673 constructs. Brachypodium and Setaria seeds were donated by the laboratories of Drs. Rick Amasino and Jean-Michel Ané (UW-Madison), respectively. The YPet fluorescent protein was kindly provided by Drs. Marisa Otegui and Makoto Yanagisawa (UW-Madison). Golden Gate vectors and plant modular cloning parts that were obtained through Addgene are courtesy of Drs. Sylvestre Marillonnet (Leibniz-Institut für Pflanzenbiochemie; Halle, Sachsen-Anhalt, Germany) and Nicola Patron (The Earlham Institute; Norwich, United Kingdom). R.Y. was supported by the postdoctoral fellowship of the Japan Society for the Promotion of Science. This work was supported by the U.S. National Science Foundation awards, PGRP-IOS-1836824 and MCB-1818040 (principal investigator, H.A.M.).

## Author contributions

H.A.M. and J.E.A. conceived the research, with the input of B.M., Y.T.K., R.Y., X.C., and M.S. The experiments were primarily planned and conducted by J.E.A. B.M. and Y.T.K. performed gene expression studies and phylogenetic analysis. X.C. and M.S. cloned grass TyrA genes and contributed to the characterization of these enzymes. R.Y. developed the UHPLC-MS method for the quantification of DHS activity and advised J.E.A. on the characterization of DHS enzymes. M.W.A. carried out the transient expression of TyrA genes in Arabidopsis protoplasts. The manuscript was written by J.E.A. and H.A.M., with input from all authors.

## Competing interests

H.A.M. and J.E.A. have a pending patent application related to the naturally deregulated DHS1b grass enzyme identified in this study. The authors declare that they have no other competing interests.
