## [Peer Review File · Nature Communications]

Coordinated regulation of the entry and exit steps of aromatic amino acid biosynthesis supports the dual lignin pathway in grassesREVIEWER COMMENTS

Reviewer #1 (Remarks to the Author):

This is an interesting manuscript linking enzyme properties and activity at the beginning of the shikimate pathway and in the biosynthesis of Tyr to ability of a plant to incorporate Tyr into lignin.

AAA biosynthesis plays an important role in generating Phe or Tyr to support polyphenol biosynthesis in plants. The overarching conclusions of this manuscript appear to be the ability to express non feedback regulated forms of these enzymes relates to the accumulation of Tyr.

There are some nice experiments looking at metabolites and including CO₂ incorporation. I have some concern the gene expression analysis does not related directly to the metabolite levels - perhaps this can be clarified or reworked so that the correlations are clear.

My main concerns with this work are related to the protein level work and the conclusions taken from experiments that may have some limitations in the way that they are conducted.

1. With all the kinetic measurements - reporting to such high levels of significant figures way beyond the precision of the data does not reflect sophisticated understanding of the data. The sfs must match data precision - these kinetic tables are absurd.
2. The kinetic tables at least in the main manuscript should be reduced to include key data. I don't see how the specific activity (here as V_{max}) adds more meaning than the cat etc. Likewise IC₅₀s when a K_i is given for a competitive model add no value.
3. A competitive model for Tyr seems reasonable - but the data should be shown - but also this should be linked to what is known about the structure of the protein. I suggest the TyrAnc - does not have any regulation domain? This is predictable from sequence and this kind of insight should be correlated to the observations.
4. I have significant concern about some of the DHS kinetics. While the basic kinetics is fine - and it is nice to see the E4P cooperativity - I am concerned about the conditions for examining feedback inhibition. It is known that these enzymes bind feedback effectors at remote sites, so allosterically - this means the model and conditions of inhibition are important. High substrate concentrations may well mask some inhibition and a range of inhibitors should be tried at variable substrate concentrations. This is not clear. Likewise combinations of inhibitors have been shown to have effect on particularly type II enzymes - combinations were not tried.
5. As with TyrA sequence can be used to predict structure and in DHS the presence of allosteric effete binding domains (and be used to predict whether combinations may be effective) - this should be used to corroborate or otherwise the findings. If there is no effectors are in enzymes that show an allosteric binding site does this mean that there has been mutational change to remove this protein level sensitivity. Such analysis would provide more insight and reassure that biochemical and physiological functioning had not been overlooked.

Reviewer #2 (Remarks to the Author):

That grasses biosynthesize phenylpropanoids from both phenylalanine and tyrosine has been known for over sixty years, but until recently it had remained a relatively under-studied aspect of plant metabolism, particularly in comparison to the numerous publications on the synthesis in dicots of phenylpropanoids from phenylalanine via PAL. Those studies that had been conducted primarily focused on the differences between PAL and PTAL. In this study, the authors have tackled a much more difficult and interesting question to gain insight into how (or even whether) grasses can synthesize sufficient levels of tyrosine to support lignin biosynthesis via PTAL. As such, this work is very important and timely, particularly given interest in biofuel production using grass species such as *Miscanthus* and switchgrass as feedstocks.

In brief, through an elegant series of experiments, the authors have tracked down the ability of grasses to make lignin from tyrosine to isoforms of DHS and TyrA that have evolved to have altered feedback regulation. They come to these conclusions following isotopic feeding experiments, in vitro enzyme characterization, and transgenic plant analysis. It is also noteworthy that they characterized enzymes from not just one species, but both sorghum and *Brachypodium*, and are thus better able to generalize their observations across grasses. In addition to gaining insight into how plants naturally regulate carbon flux toward tyrosine, the authors also identify powerful tools for redirecting AAA pathways for altered amino acid accumulation, making this manuscript of great interest in the context of synthetic biology.

Although I am generally a very tough reviewer, I can find very little in this manuscript to criticize. Indeed, after reading the work on TyrA and then DHS, I found myself wondering what would be the outcome of a combinatorial approach, thinking that the authors would probably want to save those experiments for a subsequent paper. I was delighted to see that exactly that experiment was then included in the final section of the Results. Overall, I think this is an extremely strong and comprehensive paper. And while I am complimenting the authors, I would like to applaud them for their scholarship in citing the original work by research like Arthur Neish and Neil Towers.

A few minor comments.

Pg. 7. Lines 23-25. The authors state "Feeding experiments using $^{13}\text{CO}_2$ revealed that, beyond high tyrosine production, grass species actively synthesize shikimate and phenylalanine as well (Figures 1 and 2). This (sic) data suggests that grasses altered the regulation of the upstream shikimate pathway, upstream of TyrA." This seems like a poor justification for the experiment to come. Clearly grasses actively synthesize shikimate and phenylalanine. If they did not, they would be dead. Further, the fact that they simply synthesize it does not suggest anything to do with altered regulation. This should be a simple fix. But if the word data is used, please use "...these data..." as data is plural.

Pg 8, line 31. The authors state "Arogenate incubated with HCl, which causes its hydrolyzation into phenylalanine, did not inhibit the DHS enzymes, supporting that the inhibitory compound was arogenate instead of other possible contaminants (Supplemental Figure S10)." Exactly what kind of a reaction is this conversion? The authors at least mean hydrolysis rather than hydrolyzation which I don't think is a word, but doesn't the enzyme actually catalyze a simultaneous decarboxylation and dehydration?

Pg. 9, line 30-33. The authors state “To avoid causing a strong over-expression that may not be representative of how different enzymes integrate in a plant system, BdTyrA and BdDHS were driven by RuBisCO small subunit promoters from Arabidopsis (PAtRbcS3B) and tomato (PSIRbcS3A), respectively (Supplemental Figure S7).” I’m afraid that I don’t know what “that may not be representative of how different enzymes integrate in a plant system” means. Can the authors rephrase?

Pg. 11, line 26-28. The authors state “Surprisingly, our data showed that the most highly expressed grass TyrA in the internodes, TyrA1, encodes a tyrosine-inhibited enzyme (Figure 3), which is incongruent with the highly active tyrosine synthesis in grass internodes (Figure 2).” They then go on to articulate potentially true but complicated models based on metabolite-protein interactions. Isn’t this investing rather a lot in modest differences in expression levels when the authors don’t know whether or not the qRT results actually correlate with protein levels? Furthermore, I think that many readers will be confused by the results about tyrosine-insensitivity of TyrAnc, only to have it implied in this section that those results don’t seem to matter. I think that authors are really shortchanging themselves and their own data by not making a more nuanced argument about the potential importance of the characteristics of TyrAnc.

Pg. 22. We now pay more attention to disabilities like color blindness during the preparation of figures. In that vein, I feel obliged to point out that I find many of the authors figures almost impossible to read because of my aging eyes. Even with reading glasses on and illuminating the figures with additional light from my cell phone, I could barely read some of the text in the figures, most notably the labels within the construct figures. Can they be increased in size?

Pg. 29. The capitalization of the titles in the references is irregular. Italics are also missing in some cases. Journal title abbreviations are not consistent (The Plant Cell vs. Plant Cell, periods after abbreviations sometimes there and sometimes not).

Responses to Referees' Comments:

“Coordinated Regulation of the Entry and Exit Steps of Aromatic Amino Acid Biosynthesis Supports the Dual Lignin Pathway in Grasses”

El-Azaz et al.

Reviewer #1 comments:

This is an interesting manuscript linking enzyme properties and activity at the beginning of the shikimate pathway and in the biosynthesis of Tyr to ability of a plant to incorporate Tyr into lignin.

AAA biosynthesis plays an important role in generating Phe or Tyr to support polyphenol biosynthesis in plants. The overarching conclusions of this manuscript appear to be the ability to express non feedback regulated forms of these enzymes relates to the accumulation of Tyr.

There are some nice experiments looking at metabolites and including CO₂ incorporation. I have some concern the gene expression analysis does not related directly to the metabolite levels - perhaps this can be clarified or reworked so that the correlations are clear.

Response: We reworked the results section, page 6, lines 1-2, to describe that the expression profile of TyrA1 and TyrAnc “correlates with the elevated rate of tyrosine production observed in grass internodes in the ¹³CO₂ feeding experiments (Figure 2)”. We hope that the new version makes a clearer connection between transcriptional and metabolite observations.

My main concerns with this work are related to the protein level work and the conclusions taken from experiments that may have some limitations in the way that they are conducted.

1. With all the kinetic measurements - reporting to such high levels of significant figures way beyond the precision of the data does not reflect sophisticated understanding of the data. The sfs must match data precision - these kinetic tables are absurd.

Response: The legends of Figures 3 and 4, Tables 1 and 2, the materials and methods section have been modified to clarify the number of replications used and the accuracy of the data shown. All kinetic determinations included in the manuscript are based on n=4 to 6 per datapoint, which are derived from two or more technical replicates from 2 to 3 independent experiments that were conducted using different batches of purified enzymes. Significant variation across independent experiments was observed for certain enzymes like SbTyrA2 (Supplemental Figure S6A). We decided not to exclude any of these results, as they may reflect the difference in the stability of certain TyrAs during the purification process, which might be helpful for other researchers.

2. The kinetic tables at least in the main manuscript should be reduced to include key data. I don't see how the specific activity (here as Vmax) adds more meaning than the cat etc. Likewise IC50s when a Ki is given for a competitive model add no value.

Response: The kinetic tables have been simplified to avoid redundancy of some of the data/parameters shown. Vmax values have been excluded from Tables 1 and 2 to show only the derived kcat parameter. The number of decimals has been reduced across tables 1 and 2 to make the data interpretation and comparison easier.

3. A competitive model for Tyr seems reasonable - but the data should be shown - but also this should be linked to what is known about the structure of the protein. I suggest the TyrAnc - does not have any regulation domain? This is predictable from sequence and this kind of insight should be correlated to the observations.

Response: We obtained new data showing competitive inhibition kinetics for BdTyrA1 and BdTyrAnc, which have been included in the new Supplemental Figure S7. These findings have been cited in the main text (page 7, lines 2-4). These experiments confirm that the inhibition of TyrAnc by tyrosine is competitive, which is consistent with previous reports for other TyrAs (Connely and Conn, 1983; Rippert and Matringe, 2002). Our prior studies (Scheck et al., 2017; Lopez-Nieves et al., 2021) provided the structural basis of tyrosine binding to the active site as a competitive inhibitor in legumes TyrAp, which shares a relatively high sequence identity with grass TyrAnc (~70%, versus ~50% with *Brachypodium* TyrA1 or *Arabidopsis* TyrA2). Multiple sequence alignments and *in silico* models of plant TyrAs also support that grass TyrAnc enzymes have the same primary structure than the rest of plant TyrAs. Therefore, the existence of a dedicated regulatory domain mediating an allosteric-like inhibition mechanism -like the ACT domain of plant arogenate dehydratases- seems very unlikely in this family of plant enzymes.

4. I have significant concern about some of the DHS kinetics. While the basic kinetics is fine - and it is nice to see the E4P cooperativity - I am concerned about the conditions for examining feedback inhibition. It is known that these enzymes bind feedback effectors at remote sites, so allosterically - this means the model and conditions of inhibition are important. High substrate concentrations may well mask some inhibition and a range of inhibitors should be tried at variable substrate concentrations. This is not clear. Likewise combinations of inhibitors have been shown to have effect on particularly type II enzymes - combinations were not tried.

Response: We appreciate the reviewer's assessment regarding these shortcomings. We have conducted a detailed characterization of the inhibition kinetics of BdDHS2 for arogenate and tryptophan and presented the data in new Supplemental Figure S13. The new findings show that both inhibitors cause a drastic decrease in V_{max} , but have more varied effects on the K_m/EC_{50} for the substrates, with inhibition modes that range between non-competitive, uncompetitive and mixed. Similar findings were previously reported for the inhibition of bacterial type-II DHSs by aromatic amino acids as effector molecules (Webby et al., 2010. *J Biol Chem*, 285:30567-30576). Considering these inhibition kinetics, DHS enzyme inhibition is detectable at a wide range of substrates, especially at high concentrations. Therefore, the saturating concentrations of substrate should not mask inhibition of DHS activity by the effector molecules.

In addition, we further tested the combined effects of four inhibitors—tyrosine, tryptophan, chorismate and arogenate—that impacted the activity of grass DHSs when assayed independently at a relatively high concentration of 0.5 mM. These new assays on BdDHS1a, BdDHS1b, and BdDHS2 with two or more effectors have been included in the new Supplemental Figure S14. Overall, we have observed some additive, but not synergistic, effects on BdDHS1a and BdDHS2 inhibition. BdDHS1b was not inhibited by any of the treatments tested, except for minor inhibition by chorismate, which inhibitory effect disappears when aromatic amino acids are included.

5. As with TyrA sequence can be used to predict structure and in DHS the presence of allosteric effete binding domains (and be used to predict whether combinations may be effective) - this should be used to corroborate or otherwise the findings. If there is no effectors are in enzymes that show an allosteric binding site does this mean that there has been mutational change to remove this protein level sensitivity. Such analysis would provide more insight and reassure that biochemical and physiological functioning had not been overlooked.

Response: The reviewer's comment points out a very relevant question about the mechanistic insights of DHS feedback regulation in plants, which remains poorly understood due to the lack of plant DHS structures co-crystallized with their effector molecules. The sota mutations that we recently identified in

Arabidopsis DHSs (Yokoyama et al., 2022) support that mutations in the putative allosteric binding site, away from the catalytic site, are responsible for alleviating effector-mediated feedback inhibition in Arabidopsis DHSs. However, the sequence of this effector binding site is well conserved in grass DHS1b, which this study found to be resistant to feedback inhibition: in other words, none of the mutations that cause the Arabidopsis *sota* phenotype are present in grass DHS1b, suggesting that other parts of the enzyme may be also involved in determining (in)sensitivity to feedback regulation. The use of in silico models provided limited information about the evolution of feedback regulation in plant DHSs, as there are very large differences in primary sequence between type-II bacterial enzymes (for which we do have crystal structures) and their plant counterparts. We have been working to obtain a crystal structure of Arabidopsis and Brachypodium DHSs bound to their effector molecules but have not been successful so far. We hope to be able to advance this work and report in the future.

Reviewer #2 comments:

That grasses biosynthesize phenylpropanoids from both phenylalanine and tyrosine has been known for over sixty years, but until recently it had remained a relatively under-studied aspect of plant metabolism, particularly in comparison to the numerous publications on the synthesis in dicots of phenylpropanoids from phenylalanine via PAL. Those studies that had been conducted primarily focused on the differences between PAL and PTAL. In this study, the authors have tackled a much more difficult and interesting question to gain insight into how (or even whether) grasses can synthesize sufficient levels of tyrosine to support lignin biosynthesis via PTAL. As such, this work is very important and timely, particularly given interest in biofuel production using grass species such as Miscanthus and switchgrass as feedstocks.

In brief, through an elegant series of experiments, the authors have tracked down the ability of grasses to make lignin from tyrosine to isoforms of DHS and TyrA that have evolved to have altered feedback regulation. They come to these conclusions following isotopic feeding experiments, in vitro enzyme characterization, and transgenic plant analysis. It is also noteworthy that they characterized enzymes from not just one species, but both sorghum and Brachypodium, and are thus better able to generalize their observations across grasses. In addition to gaining insight into how plants naturally regulate carbon flux toward tyrosine, the authors also identify powerful tools for redirecting AAA pathways for altered amino acid accumulation, making this manuscript of great interest in the context of synthetic biology.

Although I am generally a very tough reviewer, I can find very little in this manuscript to criticize. Indeed, after reading the work on TyrA and then DHS, I found myself wondering what would be the outcome of a combinatorial approach, thinking that the authors would probably want to save those experiments for a subsequent paper. I was delighted to see that exactly that experiment was then included in the final section of the Results. Overall, I think this is an extremely strong and comprehensive paper. And while I am complimenting the authors, I would like to applaud them for their scholarship in citing the original work by research like Arthur Neish and Neil Towers.

A few minor comments.

Pg. 7. Lines 23-25. The authors state “Feeding experiments using $^{13}\text{CO}_2$ revealed that, beyond high tyrosine production, grass species actively synthesize shikimate and phenylalanine as well (Figures 1 and 2). This (sic) data suggests that grasses altered the regulation of the upstream shikimate pathway, upstream of TyrA.” This seems like a poor justification for the experiment to come. Clearly grasses actively synthesize shikimate and phenylalanine. If they did not, they would be dead. Further, the fact that they simply synthesize it does not suggest anything to do with altered regulation. This should be a simple fix. But if the word data is used, please use “...these data...” as data is plural.

Response: We agree on the inaccurate use of the expression “actively synthesize” to summarize these findings. The expression has been replaced by “grass species also synthesize shikimate and phenylalanine at a higher rate than Arabidopsis” (page 7, lines 26-27). The title of the first section of the results has been modified accordingly. The “this data” typo has been fixed (page 7, line 28).

Pg 8, line 31. The authors state “Arogenate incubated with HCl, which causes its hydrolyzation into phenylalanine, did not inhibit the DHS enzymes, supporting that the inhibitory compound was arogenate instead of other possible contaminants (Supplemental Figure S10).” Exactly what kind of a reaction is this conversion? The authors at least mean hydrolysis rather than hydrolyzation which I don’t think is a word, but doesn’t the enzyme actually catalyze a simultaneous decarboxylation and dehydration?

Response: The conversion of arogenate into Phe under acidic conditions (Zamir et al., 1983) is analogous to the decarboxylation and dehydration catalyzed by the enzyme arogenate dehydratase on the phenylalanine pathway. The wording of this sentence has been changed to properly describe this acid-treated arogenate control.

Pg. 9, line 30-33. The authors state “To avoid causing a strong over-expression that may not be representative of how different enzymes integrate in a plant system, BdTyrA and BdDHS were driven by RuBisCO small subunit promoters from Arabidopsis (PAtrbcS3B) and tomato (PSIRbcS3A), respectively (Supplemental Figure S7).” I’m afraid that I don’t know what “that may not be representative of how different enzymes integrate in a plant system” means. Can the authors rephrase?

Response: Our intention was to emphasize the use of endogenous plant promoters rather than other popular promoters (CaMV35S) which drive very high expression levels that would rarely occur outside of the laboratory experiments. After considering the reviewers’ comment, the statement “that may not be representative of how different enzymes integrate in a plant system” has been removed to prevent any confusion.

Pg. 11, line 26-28. The authors state “Surprisingly, our data showed that the most highly expressed grass TyrA in the internodes, TyrA1, encodes a tyrosine-inhibited enzyme (Figure 3), which is incongruent with the highly active tyrosine synthesis in grass internodes (Figure 2).” They then go on to articulate potentially true but complicated models based on metabolite-protein interactions. Isn’t this investing rather a lot in modest differences in expression levels when the authors don’t know whether or not the qRT results actually correlate with protein levels? Furthermore, I think that many readers will be confused by the results about tyrosine-insensitivity of TyrAnc, only to have it implied in this section that those results don’t seem to matter. I think that authors are really shortchanging themselves and their own data by not making a more nuanced argument about the potential importance of the characteristics of TyrAnc.

Response: We have reworked this section to make a clearer discussion about how TyrA1, TyrAnc, and DHS1b may be operating to achieve the high carbon flux towards tyrosine and phenylalanine production observed in grass tissues.

Pg. 22. We now pay more attention to disabilities like color blindness during the preparation of figures. In that vein, I feel obliged to point out that I find many of the authors figures almost impossible to read because of my aging eyes. Even with reading glasses on and illuminating the figures with additional light from my cell phone, I could barely read some of the text in the figures, most notably the labels within the construct figures. Can they be increased in size?

Response: We apologize for this oversight. After converting the figures to PDFs and merging them into the manuscript, many of the fonts became smaller than what we initially planned. We have carefully revised all the figures to make them easier to read.

Pg. 29. The capitalization of the titles in the references is irregular. Italics are also missing in some cases. Journal title abbreviations are not consistent (The Plant Cell vs. Plant Cell, periods after abbreviations sometimes there and sometimes not).

Response: We also apologize for these formatting errors. The reference list has been revised to fix the formatting inconsistencies.

REVIEWERS' COMMENTS

Reviewer #1 (Remarks to the Author):

This revision has addressed the primary concerns in the original submission.

Reviewer #2 (Remarks to the Author):

I am satisfied that the authors have addressed the points I raised in my initial review.